# LowRA: Accurate and Efficient LoRA Fine-Tuning of LLMs under 2 Bits

**Zikai Zhou** [1]    **Qizheng Zhang** [1]    **Hermann Kumbong** [1]    **Kunle Olukotun** [1]

## Abstract

Fine-tuning large language models (LLMs) is increasingly costly as models scale to hundreds of billions of parameters, and even parameter-efficient fine-tuning (PEFT) methods like LoRA remain resource-intensive. We introduce LowRA, the first framework to enable LoRA fine-tuning below 2 bits per parameter with minimal performance loss. LowRA optimizes fine-grained quantization—mapping, threshold selection, and precision assignment—while leveraging efficient CUDA kernels for scalable deployment. Extensive evaluations across 4 LLMs and 4 datasets show that LowRA achieves a superior performance–precision trade-off above 2 bits and remains accurate down to 1.15 bits, reducing memory usage by up to 50%. Our results highlight the potential of ultra-low-bit LoRA fine-tuning for resource-constrained environments.

## 1. Introduction

Fine-tuning large language models (LLMs) enhances task-specific performance (Wei et al., 2021; Wang et al., 2022; Ziegler et al., 2019) and mitigates undesired behaviors such as hallucinations (Hu et al., 2024; Liu et al., 2023) and harmful responses (Bai et al., 2022; Askell et al., 2021). However, as model sizes grow—*e.g.,* LLaMA 3.1 (405B)(Dubey et al., 2024) and DeepSeek-V3 (671B)(Liu et al., 2024a)—fine-tuning costs escalate significantly (Hu et al., 2021).

Parameter-efficient fine-tuning (PEFT) techniques (Zaken et al., 2021; Hu et al., 2021; Mao et al., 2021; Liu et al., 2022a) address this by freezing core model weights and introducing small trainable modules. LoRA (Hu et al., 2021), a widely adopted PEFT method, inserts low-rank adapters to reduce computation and memory overhead. Yet, even with LoRA, fine-tuning large models can exceed single-

GPU memory limits. Quantized LoRA approaches (*e.g.,* QLoRA (Dettmers et al., 2024), LoftQ (Li et al., 2023)) alleviate this by quantizing base model weights without accuracy loss, enabling fine-tuning on standard GPUs and even mobile devices.

Despite their success, existing quantized LoRA methods are confined to 2–4 bits per parameter (Wang et al., 2024; Meng et al., 2024; Li et al., 2023), often failing at ultra-low-bit settings (below 2 bits). Enabling such fine-tuning is critical for ultra-resource-constrained environments, such as embedded systems (Shen et al., 2023; Chai et al., 2025; Zhang et al., 2024) and mobile devices (Wang et al., 2025; Tan et al., 2024). However, current methods face three fundamental limitations:

- **L1:** Focus exclusively on coarse-grained quantization of base model weights.
- **L2:** Rely on quantization formats assuming a fixed data distribution across the model weights.
- **L3:** Depend on simulated quantization with no system-level support for efficient low-bit execution.

To unlock the full potential of quantized LoRA fine-tuning below 2 bits per parameter, we introduce LowRA, an accurate and efficient framework featuring: **(1)** a novel mapping and threshold search mechanism, **(2)** a fine-grained precision assignment strategy, and **(3)** CUDA-based quantization primitives for efficient low-bit execution.

Addressing **L1** and **L2** is particularly challenging, since LoRA base weights must remain compatible with multiple adapter sets in practical deployment scenarios (Sheng et al., 2024; Ostapenko et al., 2024; Chen et al., 2024). This necessitates **a task-agnostic, yet highly adaptive, quantization approach**. Additionally, fine-grained precision assignment requires scalable, low-complexity methods to handle large parameter spaces. LowRA tackles these with a *hierarchical Integer Linear Programming (ILP)*-based precision assigner and a *weighted Lloyd-Max* quantization formulation for adaptive quantization mapping and thresholding.

We extensively evaluate LowRA across 4 LLMs and 4 tasks, benchmarking against state-of-the-art baselines. Results show that LowRA: **(1)** achieves superior performance-precision trade-offs above 2 bits while being the first method to enable accurate LoRA fine-tuning below 2 bits, **(2)**

---

[1]Department of Computer Science, Stanford University, Stanford, USA. Correspondence to: Zikai Zhou <zikai@stanford.edu>.

*Proceedings of the $42^{nd}$ International Conference on Machine Learning*, Vancouver, Canada. PMLR 267, 2025. Copyright 2025 by the author(s).

reduces memory footprint significantly during both fine-tuning and inference, and **(3)** introduces minimal overhead despite its additional components.

In summary, we make the following contributions:

- **Identifying Limitations in Quantized LoRA:** We pinpoint three key bottlenecks in existing methods and propose solutions leveraging fine-grained precision assignment and adaptive quantization mapping/thresholding.
- **Design and Implementation of LowRA:** We develop an end-to-end framework, LowRA, incorporating a mapping/threshold learner, a precision assigner, and efficient quantization primitives for scalable fine-tuning.
- **Improved Performance-Precision Trade-Off:** Our method outperforms baselines in performance-precision trade-off, enabling an average *0.86-bit reduction per parameter with negligible performance loss*.
- **Memory Efficiency Gains:** LowRA *reduces fine-tuning memory consumption by 30–50%* and enables fine-tuning and deployment at as low as *1.15 bits per parameter*.
- **Open-Source Release:** We will open-source LowRA upon publication to foster further research in ultra-low-bit LoRA fine-tuning.

This paper is organized as follows: Section 2 covers LoRA fine-tuning and introduces three key limitations of existing quantized LoRA methods. Section 3 introduces LowRA's workflow, with Sections 4, 5, and 6 detailing its design choices, mapping/threshold search, and precision assignment. Finally, Section 7 presents our evaluation results and insights. In the Appendix, we provide in-depth details on kernel designs, ablation studies, memory usage, motivating sources, related algorithms, and supporting experiments.

## 2. Background and Motivation

### 2.1. Low-Rank Adaptation (LoRA) of LLMs

Fine-tuning large language models (LLMs) enables LLM adaptation to specific tasks or domains (Wei et al., 2021; Wang et al., 2022; Ziegler et al., 2019), but updating all parameters becomes prohibitively expensive as model sizes grow. Low-Rank Adaptation (LoRA) (Hu et al., 2021) addresses this by freezing base model weights and introducing trainable low-rank adapter matrices, significantly reducing memory and compute overhead. This method has become a cornerstone of parameter-efficient fine-tuning.

### 2.2. Quantization for LoRA Fine-Tuning

Quantized LoRA fine-tuning further cuts memory usage by quantizing the base model weights without hurting performance. QLoRA (Dettmers et al., 2024) introduces a *NormalFloat* format designed to better capture parameter

distributions., while LoftQ (Li et al., 2023), PiSSA (Meng et al., 2024), and ApiQ (Liao et al., 2024) jointly optimize quantized base weights and adapter initializations under a unified objective. These advances unlock fine-tuning and deployment of LLMs on low-resource platforms like embedded devices (Shen et al., 2023; Chai et al., 2025) and mobile phones (Wang et al., 2025; Tan et al., 2024).

### 2.3. Limitations of Existing Quantized LoRA Methods

Despite these advances, existing quantized LoRA approaches face fundamental challenges that limit their efficiency, particularly in ultra-low-bit regimes.

**L1: Coarse-Grained Precision Assignment.** Most methods apply a uniform quantization precision across entire weight matrices or layers. For example, QLoRA uses uniform 4-bit precisions for all base weights, while LoftQ adopts a layerwise mixed-precision scheme (*e.g.,* higher precision for earlier layers, lower for later layers). However, our findings (§6) suggest that unlocking ultra-low-bit fine-tuning requires a finer-grained precision assignment—potentially at the sub-layer or even sub-matrix level.

**L2: Discrepancy in Data Distribution.** Quantized LoRA methods often assume a global data distribution when selecting a quantization format. For instance, QLoRA's *NormalFloat* relies on a roughly normal distribution. However, Figure 2 shows that weight distributions vary significantly across layers and even within channels. Such discrepancies suggest that group-wise or channel-wise quantization is necessary for improved accuracy.

**L3: Lack of High-Performance Quantization Primitives.** Most quantized LoRA implementations rely on simulated quantization (Li et al., 2023; Qin et al., 2024; Bai et al., 2020), as hardware support for sub-4-bit or mixed-precision operations remains limited. For example, LoftQ requires eight A100 GPUs even for smaller LLMs, making it impractical for real-world deployment. Moreover, no existing system provides optimized low-bit CUDA kernels tailored to LoRA, exacerbating inefficiencies in fine-tuning and inference (see Appendix J for related GitHub discussions).

## 3. The LowRA Framework

In this section, we provide an overview of the LowRA end-to-end workflow, illustrated in Figure 1. The process begins with the pretrained model weights *(T1)*. We feed each layer of these weights into a dedicated mapping and thresholds learner *(P1)*, which produces optimized per-output-channel mappings and thresholds, denoted *(T2)*. These mappings and thresholds, along with the pretrained weights, are then processed by a two-step ILP quantizer *(P2)* to determine the

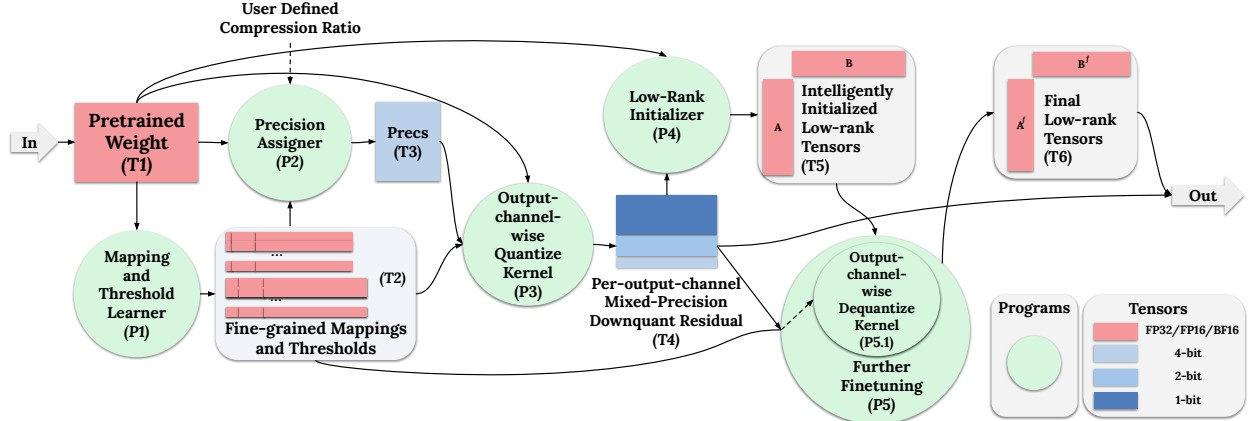

*Figure 1.* **End-to-end workflow of LowRA**. LowRA learns the mapping/threshold per output channel, assigns the precision to each output channel with a precision assigner, and supports fine-tuning workflow with Quantize and Dequantize Kernels. See section 3 for details. Limitation *L1* corresponds to components *P2* and *T3*; *L2* corresponds to *P1*, *T2*, and *P4*; and *L3* corresponds to *P3* and *P5.1*.

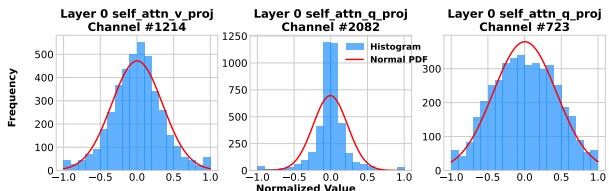

*Figure 2.* Distributions of normalized parameters in different output channels sampled from the first layer of LLaMA2-7b.

optimal precision assignments *(T3)* for each output channel.

Next, the output-channel-wise quantize kernel *(P3)*, which supports custom quantization thresholds, uses the derived thresholds *(T2)* and the assigned precision levels *(T3)* to quantize the weights. We calculate the quantization errors arising from this step and apply low-rank tensor initialization *(P4)*. Techniques for intelligent low-rank initialization include LoftQ (Li et al., 2023) and PiSSa (Meng et al., 2024) (Appendix D) which generate low-rank tensors *(T5)* designed to absorb or reduce quantization errors as possible during initialization. In our implementation, we opt for LoftQ (Li et al., 2023) as experiments show that they give better performance in the low-bit range.

The mixed-precision weights *(T4)* and initialized low-rank tensors *(T5)* feed the fine-tuning module *(P5)*, which uses an output-channel dequantize kernel *(P5.1)* to restore base weights. As in LoRA/QLoRA, base weights *(T4)* stay frozen while only the low-rank tensors *(T5)* are trained. The module outputs updated low-rank tensors *(T6)* alongside the quantized base weights and their state *(T4)*.

# 4. Discussion about Design Choices

In this section, we discuss various design choices in the LowRA framework, as well as system and hardware support.

## 4.1. Insights behind LowRA Design Choices

**Per-Output-Channel Quantization** In LLMs, linear layers often exhibit substantially more variation across output channels than across input channels. In error-sensitive layers, weight magnitudes vary markedly from one output channel to the next, producing banded patterns along the input-channel dimension that reset at each output-channel boundary (see Appendix H). As a result, grouping parameters by output channel and assigning a unique precision to each group—i.e., per-output-channel quantization—more effectively captures their diverse distributions.

**Groupwise Normalization** Each output channel may still exhibit significant internal variability even with per-output-channel quantization. To address this, groupwise normalization is often used to allow each group of elements share a separate scale. We follow QLoRA's design of using 64-element normalization scaled by the absmax (i.e., maximum absolute value) in each group (Dettmers et al., 2024).

**Data-Free Post-Training Quantization** Unlike approaches with quantization-aware training (QAT) (Esser et al., 2019; Yang et al., 2021; Jeon et al., 2024; Savarese et al., 2022), our approach adds no overhead to fine-tuning. By automatically searching for quantization mappings and thresholds, it frees users from manual tuning (Savarese et al., 2022; Zhou et al., 2023), saving both development and computation resources. Moreover, contrary to some methods that vary compression ratios over time (Savarese et al., 2022; Yang et al., 2021), LowRA maintains a consistent compression ratio, ensuring persistent memory savings during fine-tuning.

**Per-Output-Channel Thresholds and Mappings** Figure 3 visualizes the roles of thresholds and mappings in the process of quantization. Thresholds refer to the boundary points ("bin edges") that partition the continuous domain of normalized parameters into discrete intervals and thus

specific bitstring encodings. Mappings, on the other hand, specify the representative values assigned to each encoded bitstring and thus the intervals. As discussed in §2.3, fine-grained designs of quantization mappings and thresholds could lead to significantly more accurate approximation and reconstruction of parameters. LowRA allows each output channel to adopt a different combination of mappings and thresholds for more precise fine-grained quantization.

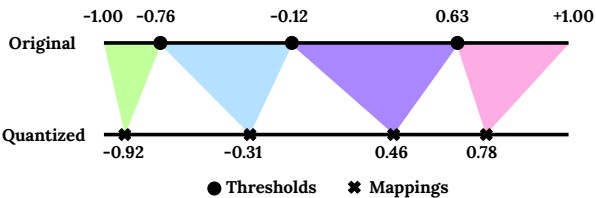

*Figure 3.* Roles of **mappings** and **thresholds** in quantization.

**Data-Free One-Shot Post-Training Quantization** Most quantization-aware training (QAT) techniques achieve higher task performance by incurring additional training overhead and learning task-specific quantization parameters (Esser et al., 2019; Yang et al., 2021; Jeon et al., 2024; Savarese et al., 2022). Similarly, many post-training quantization methods require a calibration set for quantization scheme learning (Liao et al., 2024; Hubara et al., 2021). In contrast, LowRA uses *data-free one-shot post-training quantization*, enabling reusable quantization schemes and quantized parameters, minimal hyperparameter tuning, and negligible fine-tuning overhead. This design is particularly suited to LoRA fine-tuning because: (1) Task-dependent learning is confined to the adapters, (2) LoRA base weights are often shared across multiple adapters, and (3) LoRA primarily targets resource-constrained fine-tuning scenarios.

**User-Defined Compression Ratios** Quantized LoRA methods see heavy use in tight resource settings - *e.g.,* limited-memory GPUs or on-device scenarios - where specifying a precise compression ratio is pivotal. By tailoring each parameter's bit precision, thresholds, and mappings, LowRA directly aligns compression with real-world resource budgets, ensuring feasibility and efficiency even under strict constraints. Furthermore, because LowRA fixes the ratio in a single pass, it obviates the extensive hyper-parameter tuning needed by alternative methods to find acceptable compression–accuracy trade-offs (Savarese et al., 2022).

**Using LoftQ as Low-Rank Intializer** Researchers have found that the initialization of low-rank tensors are crucial to the effectiveness of LoRA fine-tuning, especially when it comes to ultra-low-bit quantized base weight (Li et al., 2023; Meng et al., 2024; Wang et al., 2024). LoftQ (Li et al., 2023) and PiSSA (Meng et al., 2024) are two notable initialization techniques for quantized LoRA (see Appendix D for a detailed introduction). While PiSSA purports faster

convergence than LoftQ, our experiments consistently show LoftQ outperforming PiSSA. As illustrated by the sample data points in Table 1, PiSSA fails to achieve reasonable task performance at lower bit ranges. This aligns with our intuition that performing quantization rather than SVD first allows the low-rank tensors to better absorb quantization errors. Our findings also corroborate points raised in the LoftQ appendix. Since our main objective is to enable lower-precision fine-tuning and deployment, we opt to use LoftQ as our low-rank initializer. Following the recommendation from LoftQ, we use five alternating steps for initialization.

| Setup | | LLaMA-7b | | LLaMA-13b | |
|---|---|---|---|---|---|
| **Method** | **Dataset** | **2-bit** | **4-bit** | **2-bit** | **4-bit** |
| PiSSA | WikiText-2 (↓) | 1919.63 | 5.53 | 1825.68 | 5.05 |
| LoftQ | | 8.63 | 5.26 | 7.27 | 4.79 |

*Table 1.* Perplexities of PiSSA and LoftQ as initialization methods on WikiText-2. Quantization is performed at 2 bits or 4 bits per parameter. Lower values indicate better performance (↓).

**Adapting LowRA to Production Use Cases** Many production use cases, *e.g.,* batched inference in data centers, require fixed quantization mappings (Li et al., 2024; Zhao et al., 2024). LowRA can be adapted to such scenarios by keeping only the thresholds learnable, which is shown to be useful in enhancing model performance (Liu et al., 2022b). To maximize performance with task-agnostic reusable base weight, LowRA can be extended to use the same set of thresholds for multiple adapters but learn mappings for each downstream task. In other words, each adapter can be connected to the base weight together with a dedicated base weight decoding mapping for that downstream task. Nevertheless, this would demand higher development and computation costs. In our implementation and experiments, we adopt the same set of thresholds and mappings for ease of evaluation.

### 4.2. System and Hardware Support

Building on the limitations noted in §2.3, we implement practical CUDA-based primitives that support both low-bit and mixed-precision LoRA fine-tuning with maximum flexibility (details in Appendix A). Notably, the added kernel generalization incurs only negligible overhead in end-to-end inference, as quantization/dequantization constitutes a minimal portion of the total compute cost.

## 5. Mapping and Threshold Learner

In this section, we introduce the mapping and threshold learner in LowRA. Because we want the final base weights to remain task-agnostic and thus reusable across multiple adapters, we adopt a simple approach that minimizes the mean squared error in each output channel. As discussed in §4.1, one could learn separate decoding mappings for

each downstream task (or adapter set), but at a higher fine-tuning cost. We therefore propose an efficient design for the mapping/threshold learner that avoids this expense.

**Weighted Lloyd-Max Algorithm.**   We cast the problem of searching for the optimal quantization mappings and thresholds for each output channel to minimize MSE as a Weighted Lloyd-Max Problem. A detailed description of this algorithm can be found in Appendix G.

**Weighted Lloyd's for LoRA Quantization.**   As discussed in 4.1, we perform groupwise normalization to give more scale to quantization within each output channel. To recap, with groupwise normalization (Section 4.1), each block of weights is scaled by the block-wise maximum absolute value ($absmax$). Specifically, if we denote the set of original weights in a block by $\{x_i\}$ and its maximum absolute value by $absmax$, then we treat $absmax$ as a per-block weight in the Weighted Lloyd-Max algorithm. By assigning them proportionally larger weights, the algorithm "pays more attention" to those blocks and adjusts their bin thresholds and centroids accordingly. Consequently, blocks whose values have smaller magnitudes (and thus smaller $absmax$) are penalized less, striking a balance across all blocks to minimize the overall quantization error in QLoRA.

In our application of the Weighted Lloyd's algorithm to LoRA base weight quantization, we initialize the thresholds as those used by *NormalFloats* (Dettmers et al., 2024) for 2-bit and 4-bit precisions and use $0.0$ as the initial threshold for 1-bit quantization[12]. Then, at each iteration, we recompute the quantization mappings as the weighted centroids of the assigned data and recompute the thresholds as the midpoints between consecutive mapping (centroid) values[3]. We output the last-computed quantization mappings and thresholds when max iteration is reached or the MSE stops going down.

In our current implementation, we take the average of all thresholds to preserve distribution and prevent instability in the interaction with the Low-Rank Initializer.

## 6. Mixed-Precision Quantization: Channelwise Precision Assignment

In this section, we present how mixed-precision quantization assignment is conducted in LowRA. In light of the aforementioned task-agnostic requirement, a simple yet effective objective for defining this problem is the minimization of the overall Summed Square Error (SSE) considering the regular structure of transformer-based architectures (Waswani et al., 2017; Dubey et al., 2024; Touvron et al., 2023a). Such

formulation can serve as an effective proxy to retain more information for the harder-to-quantize channels in weights.

One can observe that finding the optimal mixed-precision scheme (w.r.t. SSE) can be formulated as an ILP. However, due to the large number of output channels in LLMs, a direct solver-based approach becomes computationally prohibitive. For instance, solving more than five layers of LLaMA-2-7B fails to finish within ten hours. To address this limitation, we propose a two-level ILP workflow (Figure 4, Algorithm 1) that retains the benefits of ILP-based methods while ensuring reasonable complexity.

**Notation.**   Let $N$ be the total number of output channels. Denote by $\{w^{(1)}, \ldots, w^{(K)}\}$ the distinct parameter sizes (i.e., number of parameters) appearing across channels, and let $I_k \subseteq \{1, \ldots, N\}$ be the set of channel indices whose size is $w^{(k)}$. We further define $W_k = \sum_{i \in I_k} w^{(k)}$ and $W_{\text{sum}} = \sum_{k=1}^{K} W_k$, the per-group and total parameter counts that will be used when allocating the bit-budget.

---

**Algorithm 1** Channelwise Precision Assignment

---

1: **Input**: $N$, distinct $w^{(1)}, \ldots, w^{(K)}$, partition $\{I_k\}$, MSE$(i, p)$, total budget $B_{\text{total}}$
2: **Step 1.**   Compute $W_k = \sum_{i \in I_k} w^{(k)}$, then $W_{\text{sum}} = \sum_{k=1}^{K} W_k$
3: **Step 2.** $B_k \leftarrow B_{\text{total}} \times \frac{W_k}{W_{\text{sum}}}$ for $k = 1, \ldots, K$
4: **for** $k = 1$ to $K$ **do**
5:    **Step 3.**   Cluster channels in $I_k$ (e.g. K-Means on MSE features) into $K_k$ clusters
6:    **Step 4.** *Cluster-Level ILP*: decide how many channels in each cluster get each bitwidth, subject to $B_k$
7:    **Step 5.** *Intra-Cluster ILP*: within each cluster, assign specific channels to bitwidths
8: **end for**
9: **Step 6.** Combine final bitwidths into $b_1, \ldots, b_N$
10: **Output:** $(b_1, \ldots, b_N)$, total SSE, actual bits used

---

### 6.1. Preprocessing for the Pipeline (Step 1-3)

To preprocess channels for the hierarchical ILP pipeline, we first compute each channel's MSE under 1-, 2-, and 4-bit quantization. Next, we split channels by parameter count, which in LLMs typically yields two distinct sizes (*e.g.,* 4096 and 11008 for LLaMA-2-7B). Within each group, we then apply three-dimensional K-Means clustering (based on the three computed MSE values), forming 128 clusters per group in our implementation.

### 6.2. Cluster-Level ILP (Step 4)

Formed clusters first go through the following cluster-level ILP to be assigned budgets of 1-bit, 2-bit, and 4-bit channels.

Consider $C$ clusters, each with $S_c$ channels $\left(c = 1, \ldots, C\right)$. Let $\mathcal{P} = \{1, 2, 4\}$ be the available bit-precisions[4]. For each

---

[1]Separately defined as NormalFloats lack a 1-bit representation
[2]See the initial values in Appendix F
[3]In our implementation, we set number of iterations to 2

[4]For LLaMA, restricting to 2 and 4 bits outperformed including 1 bit for bpp $\geq 2.0$, so we adopt this configuration.

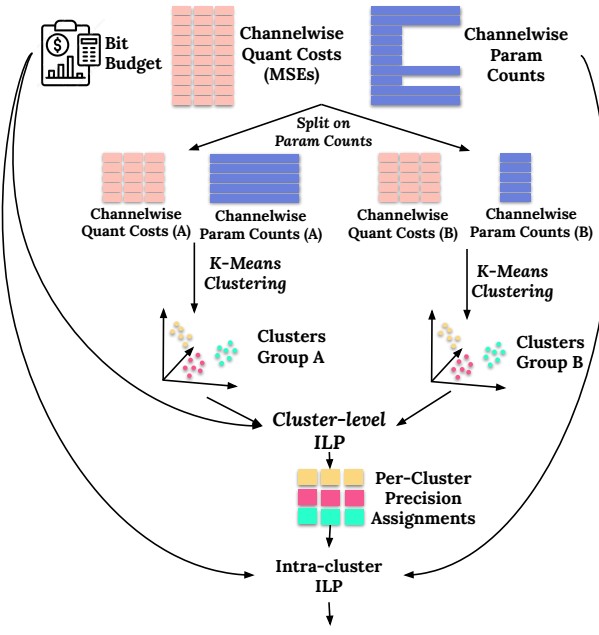

*Figure 4.* Two-step ILP-based Workflow for Channelwise Precision Assignment, with (1) channelwise clustering, (2) cluster-level ILP assignement, and (3) intra-cluster assignment, sequentially.

cluster $c$ and precision $p \in \mathcal{P}$, define:

- $\text{cost}_{c,p}$: mean quantization error (e.g., mean-squared error) *per channel* in cluster $c$ if all channels in that cluster are assigned to precision $p$, scaled by the number of weight parameters per channel in that cluster [5].
- $y_{c,p} \in \mathbb{Z}_{\geq 0}$: decision variable representing the number of channels in cluster $c$ that will be assigned precision $p$.

We define a global bit-budget $B$ (i.e., total permissible bits across all clusters). Let $\beta(p)$ be the bit-precision value (e.g., $\beta(1) = 1$, $\beta(2) = 2$, $\beta(4) = 4$). To enforce the bit budget, we multiply $\beta(p)$ by the channel parameter count $\omega_c$ for cluster $c$, and then by the number of channels $y_{c,p}$. As we do a bipartite splitting based on the number of parameters per channel, each channel in a cluster $c$ shares the same $\omega_c$.

$$
\begin{aligned}
\text{Minimize} \quad & \sum_{c=1}^{C} \sum_{p \in \mathcal{P}} \left( \text{cost}_{c,p} \right) y_{c,p} \\
\text{subject to} \quad & \sum_{p \in \mathcal{P}} y_{c,p} = S_c, \quad c = 1, \dots, C, \\
& \sum_{c=1}^{C} \sum_{p \in \mathcal{P}} \left( \beta(p)\, \omega_c \right) y_{c,p} \leq B, \\
& y_{c,p} \in \mathbb{Z}_{\geq 0}, \quad 0 \leq y_{c,p} \leq S_c.
\end{aligned}
$$

This formulation seeks to minimize the total weighted quan-

---

[5]Contrary to LoftQ's suggestion, per-layer cost weighting based on layer index proved suboptimal in our experiments.

tization error by choosing, for each cluster $c$, how many of its channels $y_{c,p}$ are assigned to each precision level $p$. The constraints ensure that every channel of a cluster is allocated exactly once, the total bits used do not exceed the overall budget $B$, and that the decision variables remain non-negative integers and do not exceed the number of channels in their respective clusters.

### 6.3. Intra-Cluster ILP (Step 5)

Once the cluster-level ILP decides how many channels $\{y_{c,p}\}$ in each cluster $c$ should be assigned to each bit precision $p$, a second ILP distributes these assignments to each channel within each cluster.

Let $S_c$ denote the total number of channels in cluster $c$. For channel $i \in \{1, \dots, S_c\}$ in cluster $c$, we define the precomputed mean-squared error at precision $p$ as $\text{MSE}(i, p) =$ (precomputed quantization error of channel $i$ at precision $p$). We define binary decision variables $x_{i,p} = 1$ if channel $i$ is assigned bit precision $p$; otherwise, $x_{i,p} = 0$.

$$
\begin{aligned}
\text{Minimize} \quad & \sum_{i=1}^{S_c} \sum_{p \in \mathcal{P}} \text{MSE}(i, p)\, x_{i,p} \\
\text{subject to} \quad & \sum_{p \in \mathcal{P}} x_{i,p} = 1 \quad \forall i \in \{1, \dots, S_c\}, \\
& \sum_{i=1}^{S_c} x_{i,p} = y_{c,p} \quad \forall p \in \mathcal{P}, \\
& x_{i,p} \in \{0, 1\} \quad \forall i \in \{1, \dots, S_c\}, \ p \in \mathcal{P}.
\end{aligned}
$$

This formulation constitutes the intra-cluster ILP. The objective minimizes the total quantization error, where $\text{MSE}(i, p)$ is the precomputed mean-squared error for channel $i$ at bit precision $p$. The first constraint ensures that each channel is assigned to exactly one precision. The second constraint enforces that the number of channels assigned to each precision $p$ matches the counts $y_{c,p}$ determined by the cluster-level ILP. Finally, the binary constraint stipulates that each decision variable $x_{i,p}$ is either 0 or 1.

This intra-cluster ILP enforces that the required number of channels (from the cluster-level ILP) is assigned to each bit precision and minimizes local MSE within the cluster.

**Efficiently Leveraging ILP Solvers.** We collect the assigned per-channel precisions in **Step 6**. By employing this two-step hierarchical approach, we harness ILP solvers' strengths while keeping computational overhead low.

## 7. Evaluation

We evaluate LowRA across four datasets spanning natural language generation, multi-turn conversation, and long-context text summarization, demonstrating that:

- **Better performance at the same precision:** LowRA

outperforms all baselines below 4-bit and matches their performance at 4-bit (§7.3).

- **Same performance at lower precision:** LowRA achieves comparable performance while reducing precision by 0.86 bits per parameter on average (§7.3).
- **First method to fine-tune LoRA under 2 bits:** LowRA enables fine-tuning down to 1.75 bits on LLaMA-2-7B, LLaMA-2-13B, and BART-large, and 1.15 bits on LLaMA-33B (§7.4).
- **Substantial memory savings:** LowRA reduces memory usage by 30–50% in fine-tuning and deployment, with minimal performance loss compared to QLoRA (§7.5).
- **Minimal overhead:** The additional one-time preprocessing costs in LowRA are negligible (Appendix E).
- **Real hardware gains:** On one GPU, 1.5-bit LowRA loads about 29 % faster and, after TTFT, delivers up to 3.4× higher throughput than 4-bit QLoRA—32.2 vs 9.4 tokens/s on LLaMA-7B and 1.4× on LLaMA-13B (§7.6).

## 7.1. Evaluation Setup

**Hardware Platform** Experiments are conducted on NVIDIA A100 GPUs (80GB memory). Each LLaMA experiment runs on a single dedicated GPU. Each BART-large experiment runs two instances concurrently on a single GPU.

**Hyperparameters** For a fair comparison, we use identical hyperparameters across all methods, consistent with QLoRA (Dettmers et al., 2024) and LoftQ (Li et al., 2023). Details on selected hyperparameters are in Appendix I.

**Language Models** We evaluate LowRA on a range of LLMs: LLaMA-2-7B, LLaMA-2-13B (Touvron et al., 2023b), BART-large (Lewis, 2019), and LLaMA-33B (Touvron et al., 2023a) (to assess ultra-low-bit scalability).

**Datasets and Evaluation Metrics** We use standard datasets across different NLP tasks: WikiText-2 (Merity et al., 2016) (language modeling, perplexity), Open-Assistant (Köpf et al., 2024) (multi-turn conversation, perplexity), XSUM (Narayan et al., 2018) (summarization, ROUGE scores), and CNN/DailyMail (Hermann et al., 2015) (summarization, ROUGE scores). Each dataset is evaluated using the standard metrics used in prior work.

## 7.2. Baselines

**QLoRA** QLoRA originally employs a fixed 4-bit quantization for pretrained LLMs and does not support fine-tuning below 4 bits. To enable sub-4-bit QLoRA experiments, we follow the adaptation introduced in LoftQ. Additionally, QLoRA directly quantizes the pretrained weights while preserving their original distribution, initializing the low-rank tensors with zeros and small Gaussian noise.

**LoftQ** LoftQ performs mixed-precision quantization (2-bit/4-bit) and jointly optimizes both quantized LLM weights

and low-rank adapter initialization. We match LoftQ's effective batch sizes but observe discrepancies with its published results due to: **(1)** reproducibility constraints – the original authors did not release experiment seeds or mixed-precision quantization hyperparameters - **(2)** hardware differences – Our experiments run on a single A100 GPU, whereas LoftQ was trained on 8 A100 GPUs with greater data parallelism - and **(3)** quantization implementation – The original LoftQ experiments rely on simulated quantization, which introduces discrepancies in quantized model weights, as noted by the research community (Liao, 2023). In contrast, we employ CUDA-kernel-based quantization and dequantization, ensuring more accurate and hardware-aligned results.

## 7.3. Analysis of Key Results

Table 2 presents the performance comparison of LowRA against QLoRA and LoftQ.

- **Better performance at the same precision:** LowRA outperforms QLoRA and LoftQ across all sub-4-bit precision levels. In particular, at challenging 2-bit quantization, LowRA achieves a perplexity reduction of: 2.21 (WikiText-2) / 1.45 (Open-Assistant) over QLoRA, and 1.76 (WikiText-2) / 1.12 (Open-Assistant) over LoftQ.
- **Same performance at lower precision:** LowRA enables fine-tuning with 0.98 bits (QLoRA) / 0.76 bits (LoftQ) fewer per parameter (on average) without performance loss. For example, 2.5-bit LowRA on WikiText-2 (LLaMA-2-7B) matches 4-bit QLoRA; 1.9-bit LowRA on Open-Assistant (LLaMA-2-7B) matches 2.5-bit LoftQ.

## 7.4. Fine-Tuning LLMs with Ultra-Low Bits

To investigate the limits of ultra-low bit LoRA fine tuning, we evaluate LowRA on two model scales: (i) LLaMA 33B and (ii) LLaMA 65B. Table 3 summarizes perplexities on WikiText 2 and OpenAssistant when training with 1.25 and 1.15 effective bits per parameter.

LowRA is the *only* approach that remains stable under these ultra-low precisions; QLoRA and LoftQ diverge at this regime. At 1.15 bits LLaMA 33B incurs merely a +0.54 absolute perplexity on WikiText 2 compared with the 1.25 bit setting, while retaining competitive quality on OpenAssistant (5.73). Strikingly, scaling the same recipe to the **65B** model yields further gains—7.49/4.97 perplexity—illustrating that LowRA **scales gracefully** with parameter count despite the extreme quantization.

## 7.5. Memory Implications

Following the analysis methodology in QLoRA (Dettmers et al., 2024), we evaluate the memory footprint of LowRA at different quantization precisions. Full visualizations are in Appendix C. Our results show that LowRA significantly reduces memory usage for both fine-tuning and in-

| Method | Bit | LLaMA-2-7B | | LLaMA-2-13B | | BART-large | |
|---|---|---|---|---|---|---|---|
| | | WikiText-2 | Open-Assistant | WikiText-2 | Open-Assistant | XSUM | CNN/DailyMail |
| | | ppl.↓/acc.↑ | ppl.↓ | ppl.↓/acc.↑ | ppl.↓ | ROUGE1↑/ROUGE2↑/ROUGEL↑ | |
| QLoRA | 4.00 | 6.22 / 0.583 | 3.52 | **4.79 / 0.628** | 3.25 | 39.07 / 16.31 / 31.09 | **41.18 / 18.32 / 27.58** |
| LoftQ | 4.00 | 5.26 / **0.613** | **3.48** | **4.79 / 0.628** | 3.23 | **40.34** / 17.06 / 31.92 | 41.12 / 18.29 / 27.54 |
| **LowRA** | 4.00 | **5.25** / 0.612 | **3.48** | **4.79 / 0.628** | **3.23** | 40.27 / **17.18 / 32.06** | 40.95 / 18.12 / 27.54 |
| QLoRA | 3.00 | 7.13 / 0.566 | 4.56 | 6.06 / 0.588 | 3.88 | 17.60 / 2.68 / 13.93 | 15.34 / 1.12 / 10.44 |
| LoftQ | 3.00 | 6.87 / 0.571 | 4.42 | 5.91 / 0.591 | 3.79 | 37.23 / 14.34 / 29.30 | 40.47 / 17.75 / 26.88 |
| **LowRA** | 3.00 | **5.84 / 0.593** | **3.87** | **5.24 / 0.611** | **3.50** | **38.84 / 15.68 / 30.57** | **40.85 / 18.12 / 27.23** |
| QLoRA | 2.50 | 8.05 / 0.546 | 5.17 | 6.84 / 0.568 | 4.36 | 15.33 / 1.97 / 12.55 | 13.68 / 1.04 / 9.99 |
| LoftQ | 2.50 | 7.72 / 0.552 | 4.98 | 6.70 / 0.572 | 4.21 | 34.48 / 12.26 / 27.05 | 39.81 / 17.19 / 26.57 |
| **LowRA** | 2.50 | **6.23 / 0.582** | **4.11** | **5.51 / 0.601** | **3.64** | **37.69 / 14.76 / 29.53** | **40.88 / 18.06 / 27.01** |
| QLoRA | 2.25 | 8.67 / 0.534 | 5.59 | 7.31 / 0.588 | 4.64 | 16.37 / 2.22 / 12.84 | 11.90 / 1.32 / 10.25 |
| LoftQ | 2.25 | 8.22 / 0.543 | 5.24 | 6.96 / 0.564 | 4.46 | 32.71 / 10.94 / 25.37 | 39.36 / 16.87 / 26.29 |
| **LowRA** | 2.25 | **6.40 / 0.578** | **4.21** | **5.66 / 0.597** | **3.73** | **37.29 / 14.36 / 29.12** | **41.01 / 18.19 / 27.23** |
| QLoRA | 2.00 | 9.17 / 0.526 | 6.07 | 7.64 / 0.551 | 5.02 | *DNC* | 4.84 / 0.00 / 4.36 |
| LoftQ | 2.00 | 8.63 / 0.536 | 5.68 | 7.27 / 0.558 | 4.75 | 31.89 / 10.18 / 24.59 | 38.88 / 16.49 / 25.85 |
| **LowRA** | 2.00 | **6.60 / 0.574** | **4.35** | **5.79 / 0.593** | **3.84** | **36.75 / 13.93 / 28.61** | **40.15 / 17.48 / 26.67** |
| QLoRA | 1.90 | – | – | – | – | – | – |
| LoftQ | 1.90 | – | – | – | – | – | – |
| **LowRA** | 1.90 | **7.13 / 0.562** | **4.94** | **6.16 / 0.583** | **4.22** | **34.05 / 11.74 / 26.49** | **39.19 / 16.84 / 26.35** |
| QLoRA | 1.80 | – | – | – | – | – | – |
| LoftQ | 1.80 | – | – | – | – | – | – |
| **LowRA** | 1.80 | **7.50 / 0.553** | **5.24** | **6.48 / 0.575** | **4.59** | **33.29 / 11.19 / 25.85** | **39.20 / 16.69 / 26.07** |
| QLoRA | 1.75 | – | – | – | – | – | – |
| LoftQ | 1.75 | – | – | – | – | – | – |
| **LowRA** | 1.75 | **7.76 / 0.548** | **5.43** | **6.65 / 0.569** | **4.76** | **33.09 / 11.05 / 25.69** | **38.54 / 16.38 / 25.99** |

*Table 2.* **Performance comparison of different methods on LLaMA-2-7B, LLaMA-2-13B, and BART-large**. "–" means this method fails to support this level of precision. "*DNC*" means fine-tuning fails to converge. LowRA (using PEFT) not only outperforms QLoRA and LoftQ in terms of performance-precision trade-off, but also enables us to fine-tune LLMs in the sub-2-bit range. LoftQ results on Bart-Large are taken as the best of two strategies: (1) layers are ordered based sheerly on layer-index and (2) encoder layers are ordered before decoder layers. See Appendix K for detailed results. Also, see Appendix B for ablation analysis.

ference, making ultra-low-bit LoRA practical on resource-constrained hardware.

For inference, reducing precision from 4-bit to 2-bit leads to 40% lower memory usage on LLaMA-2-13B and 30% on LLaMA-2-7B (Figures 8). Compressing LLaMA-33B to 1.15 or 1.25 bits achieves even greater savings, reducing the memory footprint by 50% (Figure 9).

For fine-tuning, LowRA also achieves substantial reductions. Moving from 4-bit to 2-bit precision cuts memory consumption by 30% on LLaMA-2-13B and 25% on LLaMA-2-7B (Figures 7). On LLaMA-33B, reducing precision to 1.15 or 1.25 bits per parameter leads to an estimated 45% reduction in fine-tuning memory usage, making it feasible to train larger models under stricter memory constraints.

These memory savings are particularly impactful given that 4-bit QLoRA models are already highly compressed. By pushing below 2-bit precision with minimal performance loss, LowRA enables fine-tuning and deployment

of LLMs on significantly smaller devices. For instance, a fine-tuned LLaMA-2-7B model can now be deployed on a Raspberry Pi 4 Model B (4GB RAM) (Foundation, 2019), making on-device inference feasible even in extreme resource-constrained settings. More strikingly, LowRA is the first method to enable LLaMA-33B fine-tuning on a single NVIDIA Tesla T4 (16GB VRAM) (Corporation, 2021), demonstrating its potential for democratizing large-scale LLM adaptation.

| Model | Bits | Dataset | QLoRA | LoftQ | LowRA |
|---|---|---|---|---|---|
| LLaMA 33B | 1.25 | WikiText 2 | – | – | 7.46 |
| | | OpenAssistant | – | – | 5.44 |
| | 1.15 | WikiText 2 | – | – | 8.00 |
| | | OpenAssistant | – | – | 5.73 |
| LLaMA 65B | 1.15 | WikiText 2 | – | – | 7.49 |
| | | OpenAssistant | – | – | 4.97 |

*Table 3.* Perplexity on WikiText 2 and OpenAssistant when fine tuning LLaMA-33B and LLaMA-65B with ultra low bits. "–" indicates that the method fails to converge or exceeds memory limits at the corresponding precision.

### 7.6. Inference Implications

In this section, we discuss the implications of LowRA on LLM inferences. The implications involve three primary aspects: **loading latency, time-to-first-token (TTFT), and end-to-end throughput.** We discuss each aspect below.

1. **Loading latency**: the time to move model weights into GPU memory. Loading latency is crucial for latency-sensitive, bursty (i.e., not steady-state) workloads where cold-start delays directly impact user experience and SLA compliance. Results of LLaMA-2-7B on an RTX A4000 are shown in Table 4.
2. **Time-to-first-token (TTFT)**: the latency from prompt submission to the first generated token, assuming the weights are already resident in memory. Sub-second TTFT is essential for interactive workloads because even brief stalls break the user's flow (Yao et al., 2025). Smaller or quantized models reduce KV-cache transfers slightly (Liu et al., 2024b), but the step is dominated by compute, so TTFT improvements from parameter compression are only modest.
3. **End-to-end throughput**: the sustained generation rate—tokens per second or requests per second—after the model is resident and TTFT has passed. High throughput dominates cost and user-perceived speed in long generations, batched chat sessions, and API back-ends. It scales with effective GPU compute, memory bandwidth, and parallelization depth; quantization and smaller checkpoints help, but gains come chiefly from increased arithmetic intensity and optimized kernels. Tables 5 summarize measured throughput for LLaMA-7B on an RTX 3080 and LLaMA-13B on an RTX A4000.

*Table 4.* Loading latency of LlaMA-2-7B on an RTX A4000. Using LowRA to compress model to 1.5 bits per parameter reduces loading latency by close to 30%. LoRA tensors are kept in FP32.

| Framework | Bits/Param | Latency (s) |
|---|---|---|
| QLoRA | 4 | 220.73 |
| LowRA | 4 | 218.17 |
| LowRA | 1.5 | 157.51 (–28.6%) |

**Discussion.** Across LLaMA-7B on an RTX 3080, 1.5-bit LowRA delivers a **3.42×** throughput increase over QLoRA (32.16 vs. 9.40 tokens per second). On LLaMA-13B with an RTX A4000, the same 1.5-bit configuration still yields a **1.39×** speed-up. These results confirm that sub-2-bit quantization offers real-world throughput benefits beyond the well-documented memory savings.

### 7.7. Training Overhead

We benchmarked **LowRA** at multiple bit-widths against **QLoRA (4 bit)** on an **RTX A5000** (24 GB) and an **A100** (80 GB). Runtime per iteration is shown in milliseconds; the last column reports the largest LowRA overhead versus

*Table 5.* Throughput (tokens/s) for LLaMA-2-7B and LLaMA-2-13B with on a single GPU. We experiment with a prefill length of 100 and decode length of 10. *QLoRA 4.0* refers to references results obtained with the original QLoRA implementation.

| Model | GPU | Bits | Tokens/s |
|---|---|---|---|
| LLAMA 7B | RTX 3080 | 1.5 | 32.16 |
| | | 2.0 | 30.33 |
| | | 2.5 | 24.35 |
| | | 4.0 | 9.19 |
| | | QLoRA 4.0 | 9.40 |
| LLAMA 13B | RTX A4000 | 1.5 | 16.11 |
| | | 2.0 | 14.46 |
| | | 2.5 | 13.64 |
| | | 4.0 | 12.17 |
| | | QLoRA 4.0 | 11.56 |

QLoRA. From our evaluation, LowRA adds at most **8.5 %** runtime overhead—and crucially, executes configurations that QLoRA cannot run due to out-of-memory limits.

*Table 6.* Benchmarked end-to-end training runtime (ms) of LowRA across models, hardware, and precision configurations. ***Seq.*** refers to training sequence length, ***Batch*** refers to batch number, and *QLoRA* refers to the reference training latency of QLoRA (Dettmers et al., 2024). "-" means no data/not applicable. *OOM* means "out-of-memory".

| Seq. | Batch | 1.5 bit | 2.5 bit | 3 bit | 4 bit | QLoRA | Max % |
|---|---|---|---|---|---|---|---|
| **LLAMA 7B — RTX A5000** | | | | | | | |
| 256 | 1 | 782.7 | 789.4 | – | 782.7 | 754.3 | 4.65 |
| 512 | 1 | 1291.8 | 1295.8 | – | 1291.8 | 1268.5 | 2.15 |
| 1024 | 1 | 2398.1 | 2399.0 | – | 2398.1 | 2383.7 | 0.64 |
| 256 | 2 | 1268.2 | 1274.0 | – | 1268.2 | 1247.2 | 2.15 |
| 512 | 2 | 2319.8 | 2322.0 | – | 2319.8 | 2304.8 | 0.75 |
| **LLAMA 13B — RTX A5000** | | | | | | | |
| 256 | 1 | 1494.7 | 1494.7 | 1501.2 | 1508.0 | 1444.2 | 4.42 |
| 512 | 1 | 2486.0 | 2486.0 | OOM | OOM | OOM | – |
| 256 | 2 | 2457.8 | 2457.8 | OOM | OOM | OOM | – |
| **LLAMA 7B — A100 80 GB** | | | | | | | |
| 256 | 1 | 632.1 | 632.4 | 631.9 | 639.3 | 589.1 | 8.52 |
| 512 | 1 | 1073.7 | 1074.6 | 1072.9 | 1079.5 | 1030.8 | 4.72 |
| 256 | 2 | 1056.9 | 1056.2 | 1055.9 | 1063.5 | 1014.9 | 4.79 |

## 8. Conclusion and Future Work

As LLMs grow, even parameter-efficient schemes like LoRA strain compute and memory. LowRA overcomes these limits, enabling accurate LoRA fine-tuning at <2 bits per weight through fine-grained precision assignment, adaptive quantization, and custom CUDA kernels. Across models, LowRA cuts memory by up to 50 % while matching or exceeding full-precision accuracy down to 1.15 bits, unlocking fine-tuning on mobile, embedded, and other resource-constrained devices—and charting a path toward ultra-low-bit LLM training and deployment.

## Acknowledgements

We thank our anonymous reviewers for their invaluable feedback, which significantly enhanced the quality of this paper. We also thank Yilin (Clark) Xu, Hanchen Li, Genghan Zhang, Yicheng Qian, Jungwoo Kim, and Yufei Jin for their insightful discussions. This research was supported in part by the National Science Foundation under Grant No. 2211384 (*Collaborative Research: CNS Core, Medium — A Stateful Switch Architecture for In-Network Compute*) and by the Naval Surface Warfare Center under Agreement No. N00164-23-9-G057-01 (*Project 1 — Energy-Efficient and Scalable AI Hardware Systems through Heterogeneous Integration of Specialized Chiplets*). The authors thank both sponsors for their generous support. Hermann Kumbong gratefully acknowledges support from the Stanford Knight-Hennessy Scholars Program[6]. Zikai Zhou is supported by the Stanford School of Engineering Fellowship, generously endowed by the Enlight Foundation[7].

## Impact Statement

This paper contributes to the advancement of Machine Learning. While our work may have various societal implications, none require specific emphasis. Additionally, this research does not raise any ethical concerns.

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

# A. LowRA System Support for Low-Bit Fine-Grained LoRA Fine-tuning

In this appendix, we provide an overview of the CUDA-based system support we built for supporting low-bit fine-grained quantization and dequantization for LoRA fine-tuning. We specifically introduce the **Quantize** and **Dequantize** Kernels for low-bit fine-grained LoRA fine-tuning. We integrate this into the *bitsandbytes*[8] library for usability.

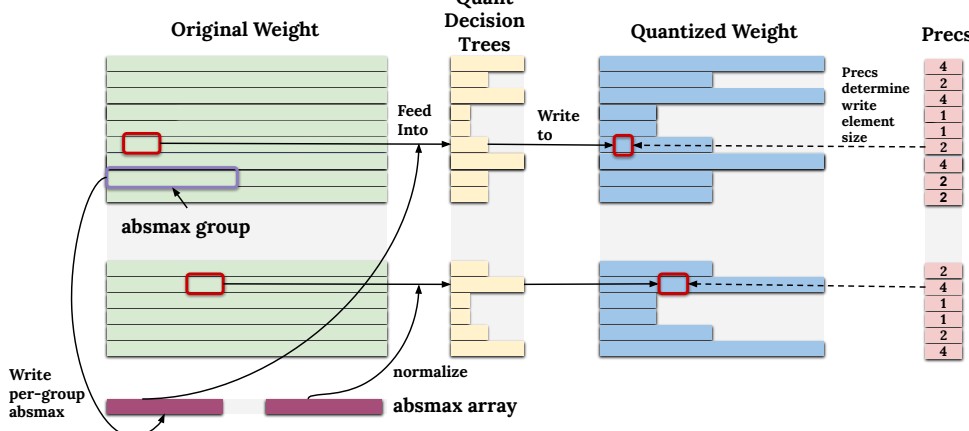

*Figure 5.* Overview of Kernel for Low-Bit Fine-Grained **Quantization**. Indexing logic omitted for simplicity.

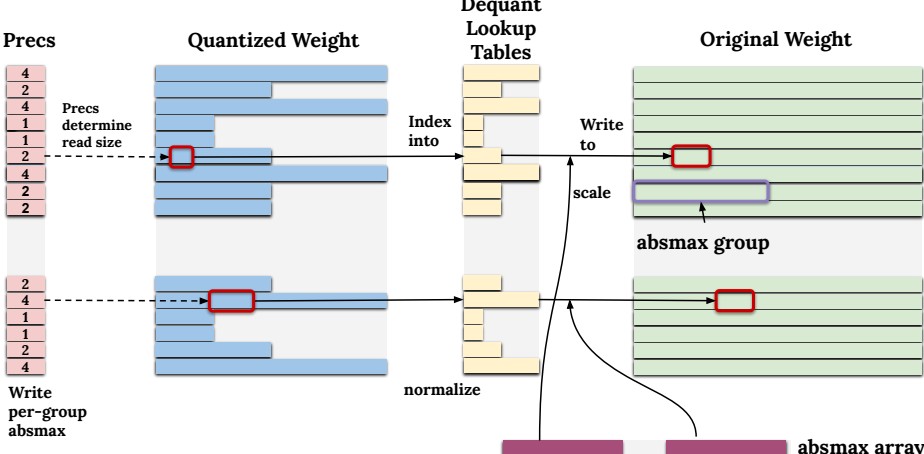

*Figure 6.* Overview of Kernel for Low-Bit Fine-Grained **Dequantization**. Indexing logic omitted for simplicity.

## A.1. Quantization (Figure 5)

During the quantization process, within the kernel, each block of data from the weight tensor is loaded, the block's maximum magnitude is computed, stored, and used for normalization, then each value is quantized according to its channel's bit-precision (defined by precs) and channel-specific decision boundaries (in the form of arrays of decision trees), and finally the packed quantized results are stored in the output buffer at the corresponding offset.

## A.2. Dequantization (Figure 6)

During the dequantization process, within the kernel, each thread calculates its offset into the quantized buffer A based on the channel's bit-precision (precs), then loads the relevant packed bytes (qvals). It uses absmax for the current block to rescale the dequantized values, which are obtained by applying a per-output-channel look-up table on each packed quantized index. Depending on the precision (1, 2, or 4 bits), the kernel unpacks the bits from qvals, looks up the corresponding float value in one of the channel-specific look-up table, multiplies by the saved group-wise absmax, and finally writes the results back into the output tensor.

---

[8]https://github.com/bitsandbytes-foundation/bitsandbytes

## B. Ablation Studies

In this appendix, we perform ablation studies on the different components of LowRA. In particular, we compare the fine-tuning results with QLoRA, LoftQ, ours with only the precision assigner, and ours with both the precision assigner and the mapping/threshold searcher. Table 7 reports results of Bart-Large on XSUM and CNN/DailyMail. Table 8 reports results of Llama-2-7B on Wikitext2.

| Technique | Metric | XSUM | | | | | CNN/DailyMail | | | | |
|---|---|---|---|---|---|---|---|---|---|---|---|
| | | 2 | 2.25 | 2.5 | 3 | 4 | 2 | 2.25 | 2.5 | 3 | 4 |
| QLoRA | ROUGE1↑ | DNC | 16.3727 | 15.3282 | 17.6011 | 39.0742 | 4.8420 | 11.8987 | 13.6767 | 15.3374 | 41.1846 |
| | ROUGE2↑ | DNC | 2.2212 | 1.9712 | 2.6801 | 16.3124 | 0.0040 | 1.3188 | 1.0364 | 1.1239 | 18.3249 |
| | ROUGEL↑ | DNC | 12.8367 | 12.5547 | 13.9294 | 31.0891 | 4.3608 | 10.2535 | 9.9929 | 10.4375 | 27.5773 |
| LoftQ | ROUGE1↑ | 31.8941 | 32.6153 | 33.7645 | 36.0222 | 40.3429 | 38.8866 | 39.3648 | 39.8138 | 40.4684 | 41.1247 |
| | ROUGE2↑ | 10.1775 | 10.7005 | 11.7335 | 13.4883 | 17.0615 | 16.4935 | 16.8711 | 17.1934 | 17.7529 | 18.2853 |
| | ROUGEL↑ | 24.5908 | 25.1533 | 26.2388 | 28.1558 | 31.9186 | 25.8534 | 26.2877 | 26.5726 | 26.8812 | 27.5372 |
| Ours, PA Only | ROUGE1↑ | 31.8941 | 36.4856 | 37.2861 | 38.1543 | 40.3429 | 38.8866 | 40.3669 | 40.6187 | 41.1820 | 41.1247 |
| | ROUGE2↑ | 10.1775 | 13.6310 | 14.3775 | 15.1856 | 17.0615 | 16.4935 | 17.6272 | 17.8600 | 18.3966 | 18.2853 |
| | ROUGEL↑ | 24.5908 | 28.4593 | 29.2036 | 29.9965 | 31.9186 | 25.8534 | 26.7442 | 26.9107 | 27.4180 | 27.5372 |
| Ours, PA + MTSearch | ROUGE1↑ | 36.7454 | 37.2915 | 37.6897 | 38.8396 | 40.2669 | 40.1489 | 41.0133 | 40.8819 | 40.8470 | 40.9493 |
| | ROUGE2↑ | 13.9324 | 14.3641 | 14.7587 | 15.6822 | 17.1800 | 17.4843 | 18.1898 | 18.0609 | 18.1191 | 18.1223 |
| | ROUGEL↑ | 28.6133 | 29.1175 | 29.5275 | 30.5712 | 32.0614 | 26.6717 | 27.2268 | 27.0113 | 27.2309 | 27.5392 |

*Table 7.* Comparison of ROUGE scores with **Bart-Large** on **XSUM** and **CNN/DailyMail** under different techniques and combinations of techniques. **PA** refers to the two-level precision assigner. **MTSearch** refers to mapping and threshold search.

| Technique | Metric | Wikitext2 | | | | | | | | |
|---|---|---|---|---|---|---|---|---|---|---|
| | | 1.5 | 1.75 | 1.8 | 1.9 | 2 | 2.25 | 2.5 | 3 | 4 |
| QLoRA | Perplexity | - | - | - | - | 9.17 | 8.67 | 8.05 | 7.13 | 6.22 |
| | Accuracy | - | - | - | - | 0.526 | 0.534 | 0.546 | 0.566 | 0.583 |
| LoftQ | Perlexity | - | - | - | - | 8.63 | 8.22 | 7.72 | 6.87 | 5.26 |
| | Accuracy | - | - | - | - | 0.536 | 0.543 | 0.552 | 0.571 | 0.613 |
| PA Only | Perplexity | ND | ND | ND | ND | 8.63 | 8.22 | 7.75 | 6.75 | 5.26 |
| | Accuracy | ND | ND | ND | ND | 0.536 | 0.542 | 0.550 | 0.570 | 0.613 |
| PA+MTSearch | Perplexity | 9.38 | 7.76 | 7.50 | 7.13 | 6.60 | 6.40 | 6.23 | 5.84 | 5.25 |
| | Accuracy | 0.521 | 0.548 | 0.553 | 0.562 | 0.574 | 0.578 | 0.582 | 0.593 | 0.6123 |

*Table 8.* Comparison of perplexity and accuracy of **Llama2-7B** on the **Wikitext2** dataset under different techniques and combinations of techniques. "ND" indicates no data, "-" indicates the techniques fail to support these precisions. **PA** refers to the two-level precision assigner. **MTSearch** refers to mapping and threshold search.

We observe generally that the precision assigner gives a significant advantage for Bart-Large on summarization tasks (Table 7), whereas the mapping/threshold searcher yields significant gains for Llama-2-7B on Wikitext2 (Table 8). Interestingly, the precision assigner only yields minimal advantage over LoftQ for Llama-2-7B on Wikitext2.

In terms of applying the mapping/threshold searcher to Bart-Large (Table 7), we see a less significant gain in the 2.5-4.0 precision range in comparison to the gains in the 2.0-2.5 precision range. This is in line with our intuition as at higher precision there are more mappings and thresholds for capturing distributions during quantization and dequantization; thereby, the tolerance for a less accurate combination of mappings and thresholds is higher.

We encourage future works to build upon LowRA and study the optimal precision assignment and mapping/threshold search

algorithms for different types of architectures (*i.e.,* encoder-decoder, encoder-only, decoder-only). We will provide easy integration to new advances in our open-sourced repository.

# C. Memory Requirements

In this appendix, we provide a detailed breakdown of memory footprints. Figure 7 shows the ***fine-tuning*** memory footprint decompositions for **LLaMA-2-7B** and **LLaMA-2-13B**. Figure 8 shows the ***inference*** memory decomposition for **LLaMA-2-7B** and **LLaMA-2-13B**. Figure 9 shows ***both inference and fine-tuning*** memory footprint decomposition for **LLaMA-33B**. For fine-tuning, we follow QLoRA's setup of using a batch size of 1 and sequence length of 512. Number labels on the bars are in MegaBytes (MB). Estimations are linear layer only (not attention).

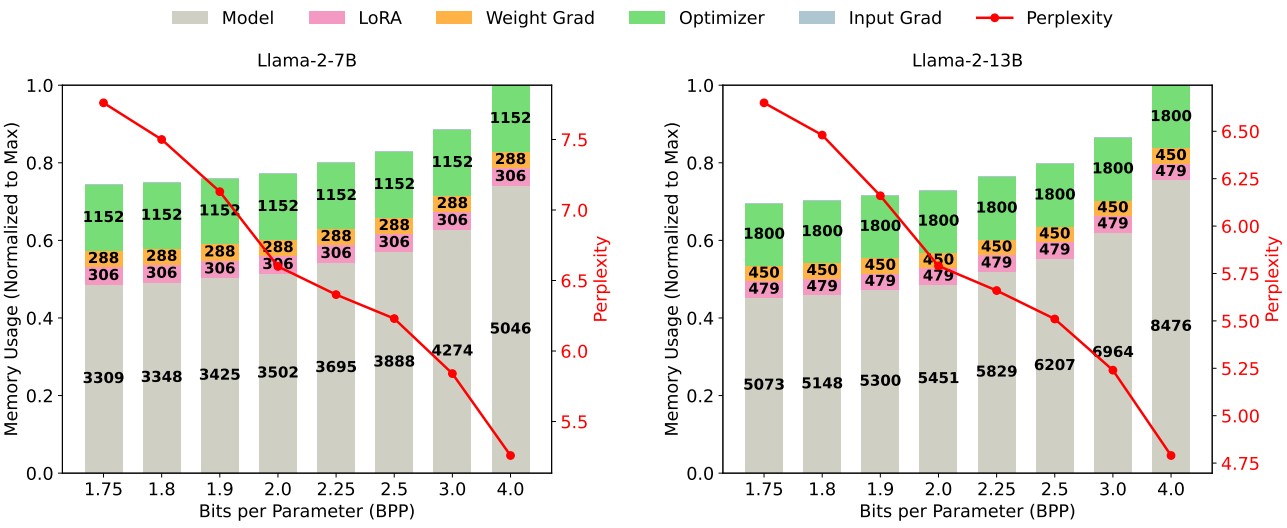

Figure 7. Decomposition of ***fine-tuning*** memory footprint for LLaMA-2 7B and 13B under different bits per parameter.

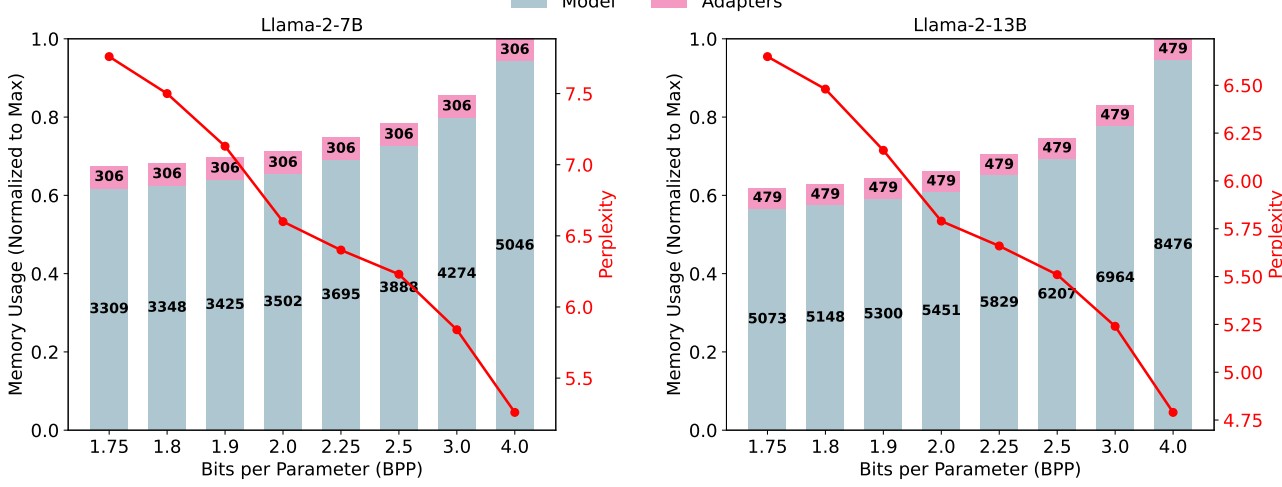

Figure 8. Decomposition of ***inference*** memory footprint for LLaMA-2 7B and 13B under different bits per parameter.

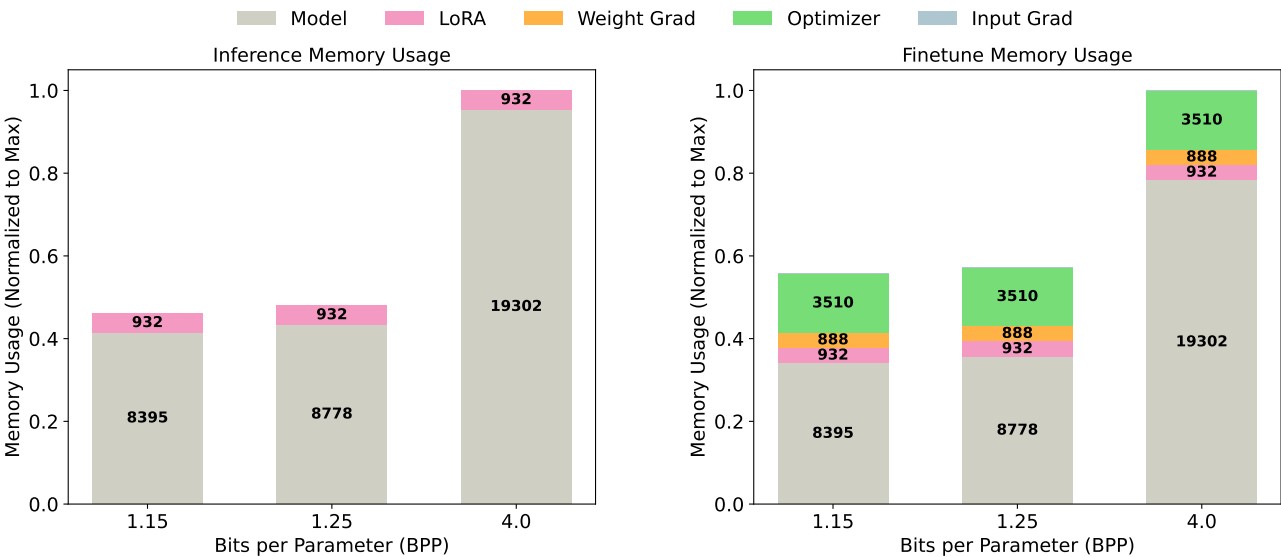

*Figure 9.* Decomposition of memory footprint for LLaMA-33B under different bits per parameter.

## D. Low-Rank Initializers: LoftQ vs PiSSA

Both LoftQ (Li et al., 2023) and PiSSa (Meng et al., 2024) use an iterative two-step process to enhance LoRA fine-tuning by exploiting low-rank structure and quantizing the residual. In each iteration:

1. **Low-Rank Decomposition:** A singular value decomposition (SVD) of the current weight (or an updated version of it) is performed to factor out a low-rank approximation.

2. **Residual Quantization:** The remaining component (i.e., the difference between the original weight and the low-rank approximation) is quantized to preserve overall model capacity with fewer bits.

The key distinction lies in how they initialize this process:

- **LoftQ** first quantizes the residual, which at the beginning is simply the full base weight. Thus, it "initializes" by treating the entire unmodified weight as a residual to be quantized and only then proceeds with the low-rank factorization in subsequent iterations.

- **PiSSA** starts by performing SVD on the unquantized base weight, extracting a low-rank representation before any quantization. Only after factoring out the low-rank component does PiSSa quantize the remaining residual.

Below (Table 9) are the experimental results covering the full 2.0-to-4.0 range comparing these two initialization techniques. We adopt LoftQ's mixed precision scheme for intermediate precision targets.

| Setup | | LLaMA-7b | | | | | LLaMA-13b | | | | |
|---|---|---|---|---|---|---|---|---|---|---|---|
| Method | Dataset | 2.0 | 2.25 | 2.5 | 3.0 | 4.0 | 2.0 | 2.25 | 2.5 | 3.0 | 4.0 |
| PiSSA | WikiText-2 ($\downarrow$) | 1919.63 | 795.88 | 1938.33 | 1397.26 | 5.53 | 1825.68 | 1822.62 | 1769.34 | 1549.48 | 5.05 |
| LoftQ | | 8.63 | 8.22 | 7.72 | 6.87 | 5.26 | 7.27 | 6.96 | 5.7 | 5.91 | 4.79 |

*Table 9.* Perplexities of PiSSA and LoftQ as initialization methods on WikiText-2 covering full range from 2.0 to 4.0. Lower values indicate better performance ($\downarrow$).

# E. Overheads of Mapping/Thresholds Searcher and Precision Assigner

In this appendix, we report the overhead incurred by LowRA's newly added components. Note that LowRA's mapping/threshold learner and precision learner can both be done offline. The former only needs to be run for each model architecture, while the latter only needs to be run for each combination of model architecture and user-specified precision requirement.

**Mapping/Threshold Learner Overhead** We timed the overhead of our mapping/threshold learner on a single NVIDIA A100 SXM GPU with GPU memory. For each precision, we ran 2 iterations of the Lloyd-Max algorithm, which we found sufficient to push down the MSE and enhance task performance. From the average of 10 runs, each run on Bart-Large, LLaMA2-7B, and LLaMA2-13B takes 111.46, 309.87, and 464.92 seconds, respectively.

**Solver Overhead** We build our two-level ILP pipeline using the opensourced Coin-Or Branch and Cut (CBC) (Saltzman, 2002) solver via the Python-based modeling library PuLP (Mitchell et al., 2011). The experiments were run on a server with 2x Intel Xeon Gold 6342 CPUs. The solver uses 8 threads in parallel. We timed the overhead of running this two-level ILP pipeline for each combination of model architecture and precision requirement. From the average of 10 runs, each run on Bart-Large, LLaMA2-7B, and LLaMA2-13B takes 48.99 seconds, 319.13 seconds, and 665.93 seconds, respectively. Note that an exact (i.e., one-step) solver for the same problem would fail to finish in a reasonable amount of time.

# F. Initialization of the Mapping and Threshold Learner

We attach the mappings and thresholds we use for weighted Lloyd-Max algorithm in Listing 1.

```
mappings_4bit_init = torch.tensor(
    [[-1.0, -0.6961928, -0.5250731, -0.3949175, -0.28444138,
      -0.18477343, -0.09105, 0.0, 0.0795803, 0.1609302,
      0.2461123, 0.33791524, 0.44070983, 0.562617, 0.72295684, 1.0]],
    dtype=torch.float32,
    device=device
).repeat(nchannels, 1)

thresholds_2bit_init = torch.tensor(
    [[-0.5, 0.16895762, 0.66895762]],
    dtype=torch.float32,
    device=device
).repeat(nchannels, 1)

mappings_2bit_init = torch.tensor(
    [[-1.0, 0.0, 0.3379, 1.0]],
    dtype=torch.float32,
    device=device
).repeat(nchannels, 1)

thresholds_1bit_init = torch.tensor(
    [[0.0]],
    dtype=torch.float32,
    device=device
).repeat(nchannels, 1)

mappings_1bit_init = torch.tensor(
    [[-1.0, 1.0]],
    dtype=torch.float32,
    device=device
).repeat(nchannels, 1)
```

Listing 1: Initializations for bit-precision mappings and thresholds.

# G. Weighted Lloyd-Max Algorithm

In this subsection, we briefly introduce the Weighted Lloyd-Max Algorithm (Lloyd, 1982; Max, 1960).

Let $\{x_i\}_{i=1}^N \subset \mathbb{R}^d$ be data points with corresponding weights $\{w_i\}_{i=1}^N$, $w_i > 0$. We seek to find $K$ cluster centers $\{y_j\}_{j=1}^K \subset \mathbb{R}^d$ minimizing the weighted mean-squared error:

$$\min_{\{c(i)\},\{y_j\}} \sum_{i=1}^N w_i \left\| x_i - y_{c(i)} \right\|^2,$$

where $c(i) \in \{1, \ldots, K\}$ is the cluster index assigned to $x_i$. The weighted Lloyd's algorithm alternates between:

**1. Assignment (E-step):** Assign each data point $x_i$ to the cluster center closest in Euclidean distance:

$$c(i) \leftarrow \underset{1 \leq j \leq K}{\arg \min} \left\| x_i - y_j \right\|^2. \tag{E-step}$$

**2. Update (M-step):** Recompute each cluster center $y_j$ as the weighted centroid of the points assigned to it:

$$y_j \leftarrow \frac{\sum_{i:c(i)=j} w_i\, x_i}{\sum_{i:c(i)=j} w_i}. \tag{M-step}$$

These steps are repeated until convergence or until a stopping criterion (e.g., a maximum number of iterations) is met.

## H. Patterns of Parameter Values in Large Language Models

We randomly sample 200×200 blocks and plot their distributions as heatmaps. In this appendix, we present representative parameter distributions within each type of tensor in Figure 10 (note that there are two parts). Generally, we observe banded patterns across input channels and separated by output channels in Self-Attention K-Projection (Figure 10a), Self-Attention Q-Projection (Figure 10b), and Self-Attention V-Projection layers (Figure 10c), which have been shown to be more sensitive to quantization errors. For MLP Gate-Projection layers (Figure 10f), we observe some similar patterns, although not as significant. Self-Attention O-Projection layers (Figure 10d), MLP Up-Projection layers (Figure 10e), and MLP Down-Projection layers (Figure 10g) have distributions that are mostly evenly spread out.

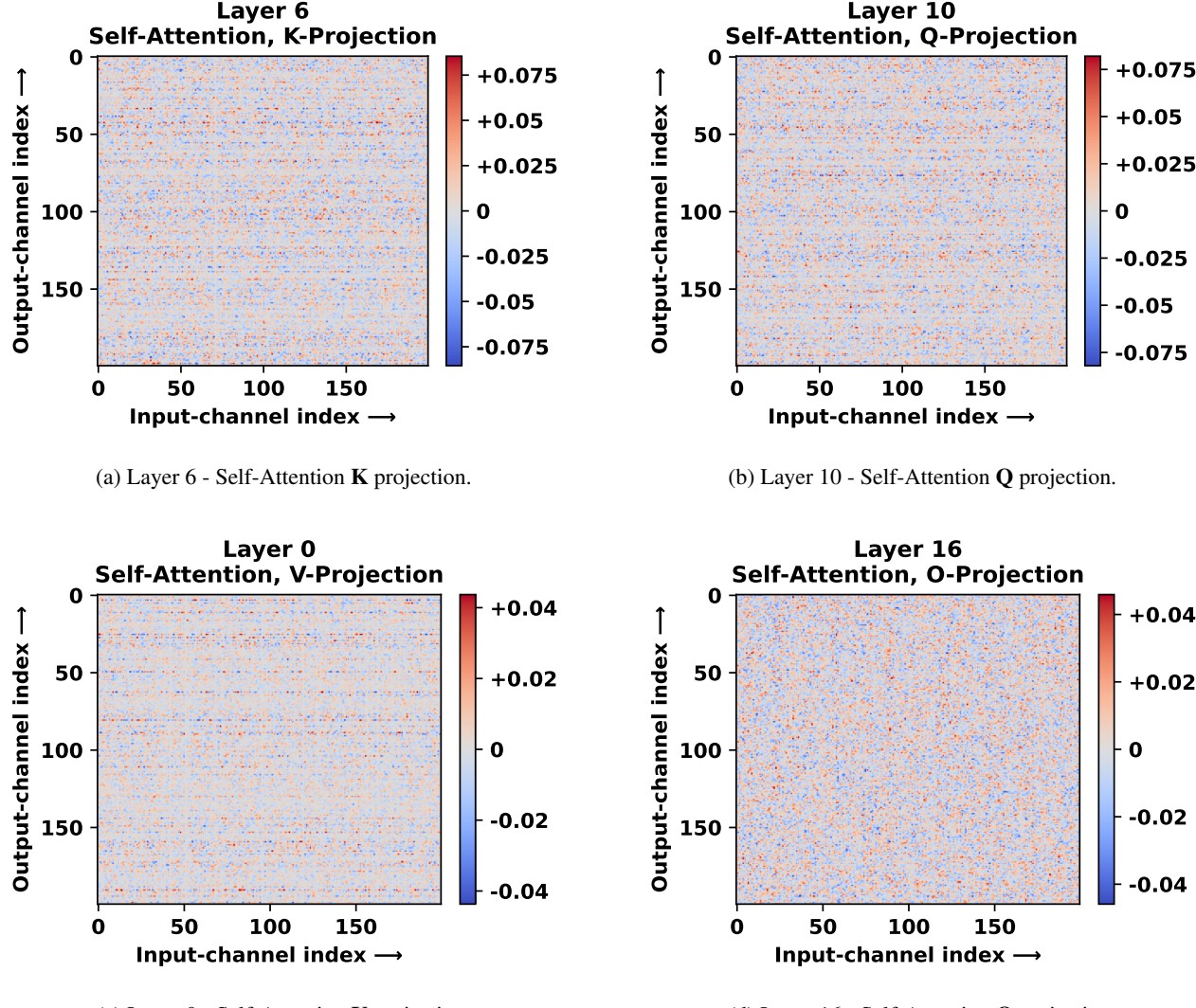

(a) Layer 6 - Self-Attention **K** projection.

(b) Layer 10 - Self-Attention **Q** projection.

(c) Layer 0 - Self-Attention **V** projection.

(d) Layer 16 - Self-Attention **O** projection.

*Figure 10.* Sampled 200×200 parameter value heatmaps from different tensors from different layers in LLaMA2-7B (1/2).

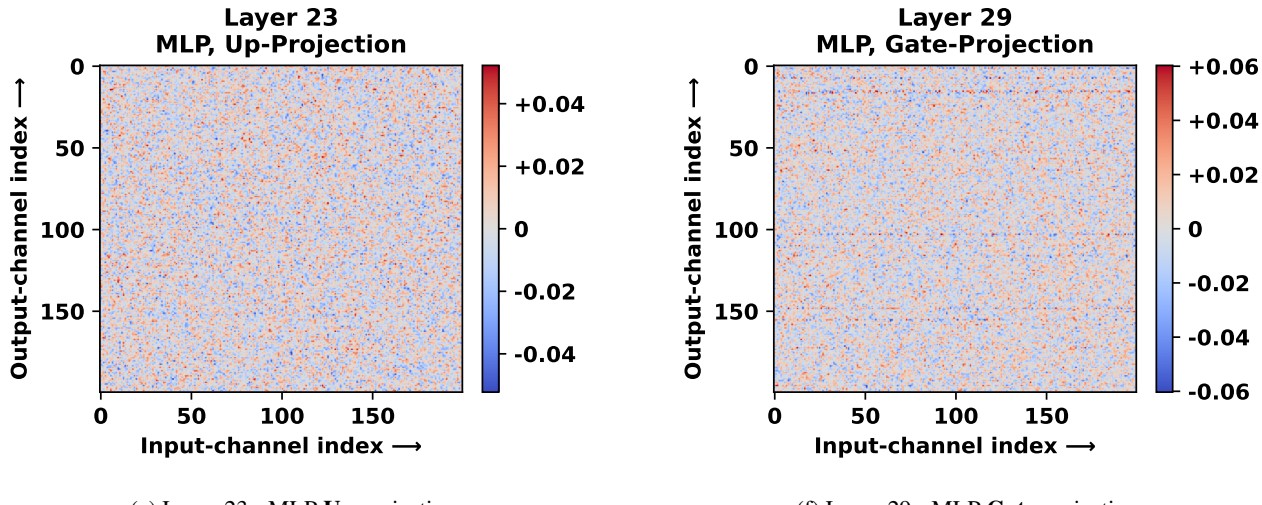

(e) Layer 23 - MLP **Up** projection.

(f) Layer 29 - MLP **Gate** projection.

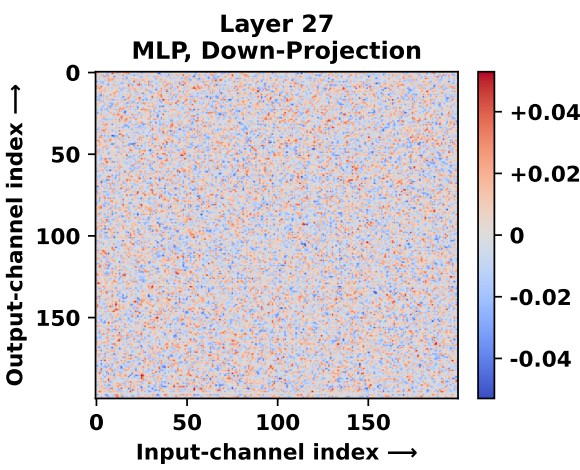

(g) Layer 27 - MLP **Down** projection.

*Figure 10.* Sampled 200×200 parameter value heatmaps from different tensors from different layers in LLaMA2-7B (2/2).

# I. Experiment Hyperparamter Setup

In this appendix, we lay out the hyperparameters used for different experiments.

| Hyperparameter | Value |
|---|---|
| model_name_or_path | meta-llama/Llama-2-7b-hf |
| data_seed | 42 |
| evaluation_strategy | steps |
| eval_dataset_size | 1024 |
| max_eval_samples | 1000 |
| per_device_eval_batch_size | 4 |
| dataloader_num_workers | 3 |
| lora_r | 64 |
| lora_alpha | 64 |
| lora_modules | all |
| bf16 | True |
| warmup_ratio | 0.03 |
| lr_scheduler_type | cosine |
| gradient_checkpointing | True |
| dataset | wikitext |
| dataset_config | wikitext-2-raw-v1 |
| per_device_train_batch_size | 16 |
| gradient_accumulation_steps | 4 |
| max_steps | 126 |
| eval_steps | 20 |
| learning_rate | 0.0003 |
| adam_beta2 | 0.999 |
| max_grad_norm | 0.3 |
| weight_decay | 0.1 |
| seed | 0 |
| block_size | 1024 |

*Table 10.* Hyperparameters used for all LLaMA experiments on Wikitext-2.

| Hyperparameter | Value |
|---|---|
| learning_rate | 1e-4 |
| seed | 11 |
| dataset_name | cnn_dailymail |
| dataset_config | "3.0.0" |
| pad_to_max_length | True |
| max_source_length | 512 |
| num_train_epochs | 15 |
| per_device_train_batch_size | 8 |
| per_device_eval_batch_size | 32 |
| gradient_accumulation_steps | 32 |
| model_name_or_path | facebook/bart-large |
| evaluation_strategy | epoch |
| predict_with_generate | True |

*Table 11.* Hyperparameters for fine-tuning Bart-Large on CNN/DailyMail.

| Hyperparameter | Value |
|---|---|
| learning_rate | 1e-4 |
| seed | 11 |
| dataset_name | xsum |
| dataset_config | "3.0.0" |
| pad_to_max_length | True |
| max_source_length | 512 |
| num_train_epochs | 25 |
| per_device_train_batch_size | 4 |
| per_device_eval_batch_size | 32 |
| gradient_accumulation_steps | 32 |
| model_name_or_path | facebook/bart-large |
| evaluation_strategy | epoch |

*Table 12.* Hyperparameters for fine-tuning Bart-Large on XSUM.

| Hyperparameter | Value |
|---|---|
| model_name_or_path | meta-llama/Llama-2-7b-hf |
| data_seed | 42 |
| evaluation_strategy | steps |
| eval_dataset_size | 1024 |
| max_eval_samples | 500 |
| per_device_eval_batch_size | 1 |
| max_new_tokens | 32 |
| dataloader_num_workers | 3 |
| group_by_length | True |
| logging_strategy | steps |
| remove_unused_columns | False |
| lora_r | 64 |
| lora_alpha | 64 |
| lora_modules | all |
| bf16 | True |
| warmup_ratio | 0.03 |
| lr_scheduler_type | constant |
| gradient_checkpointing | True |
| dataset | oasst1 |
| source_max_len | 16 |
| target_max_len | 512 |
| per_device_train_batch_size | 4 |
| gradient_accumulation_steps | 4 |
| max_steps | 1875 |
| eval_steps | 200 |
| learning_rate | 0.0002 |
| adam_beta2 | 0.999 |
| max_grad_norm | 0.3 |
| lora_dropout | 0.1 |
| weight_decay | 0.0 |
| seed | 0 |

*Table 13.* Hyperparameters used for all Llama experiments on Open Assistant (oasst1).

## J. Github Issues Related To The Lack of A Practical Quantization Primitive

- `https://github.com/yxli2123/LoftQ/issues/1`: The use of NF fake quantization in LoftQ

- `https://github.com/yxli2123/LoftQ/issues/7`: Discrepancy between real model weights and expected model weights due to fake quantization in LoftQ

- `https://github.com/yxli2123/LoftQ/issues/23`: The use of NF fake quantization in LoftQ

- `https://github.com/yxli2123/LoftQ/issues/36`: Unrealized GPU memory saving due to fake quantization in LoftQ

- `https://github.com/GraphPKU/PiSSA/issues/30`: Unrealized LLM model size reduction due to fake quantization in PiSSA

## K. Detailed LoftQ Results on Bart-Large

| Setup | | Bart-Large | | | | |
|---|---|---|---|---|---|---|
| Method | Dataset | 2.0 | 2.25 | 2.5 | 3.0 | 4.0 |
| Layer Index Encoder First | XSUM | 31.89/10.18/24.59 | 32.62/10.70/25.15 32.72/10.94/25.38 | 33.76/11.73/26.24 34.49/12.26/27.05 | 36.02/13.49/28.16 37.23/14.34/29.31 | 40.34/17.06/31.92 |
| Layer Index Encoder First | CNN/DailyMail | 38.89/16.49/25.85 | 39.36/16.87/26.29 39.27/16.84/26.26 | 39.81/17.19/26.57 39.53/17.05/26.45 | 40.47/17.75/26.88 39.89/17.36/26.73 | 41.12/18.29/27.54 |

*Table 14.* Rouge-1/Rouge-2/Rouge-L of Bart-Large fine-tuned with LoftQ. Higher scores indicate better task performance.

