# OpenReview forum: "LowRA: Accurate and Efficient LoRA Fine-Tuning of LLMs under 2 Bits"
_ICML.cc/2025/Conference — ICML 2025 poster_

### Official Review · Reviewer_BPFc · 2025-03-03

**Overall Recommendation:** 4

**Summary:**

The authors of this paper tackle the important problem of LLM quantization, enabling fine-tuning below 2 bits per parameter with minimal performance loss. This is achieved through the proposed LowRA framework, which addresses three key challenges in quantized LoRA fine-tuning: coarse-grained precision assignment, discrepancy in data distribution, and lack of high-performance quantization primitives.

The LowRA framework effectively resolves these challenges via different techniques such as per-output-channel quantization, groupwise normalization, data-free post-training quantization, per-output-channel thresholds and mappings, data-free one-shot post-training quantization, and user-defined compression ratios. Furthermore, the authors provide practical CUDA-based primitives for seamless implementation.

The paper's experimental results demonstrate that LowRA outperforms baselines at the same precision and achieves similar performance to baselines at lower precisions. Furthermore LowRA results in substantial memory savings. This makes LowRA a promising solution for embedded systems and mobile devices.

# Update after rebuttal

As mentioned in the comment below, I have decided not to update my scores.

**Claims And Evidence:**

Yes, the claims made in this paper are supported through clear and convincing evidence. In particular the authors performed experiments using four LLMs and four tasks. Furthermore, the authors provided ablation studies of the different components in LowRA which is an important addition to the study.

**Essential References Not Discussed:**

N/A

**Experimental Designs Or Analyses:**

The main experimental results are highlighted in Table 2. I have the following comments/questions for the authors here:

* LoftQ outperforms LowRA in terms of accuracy for 4 Bit LLaMA-2-7B in WikiText-2 and in terms of ROUGE1 for 4 Bit BART-large for XSUM. I might have missed this but am curious to know the authors thoughts in this regard.
* 4Bit LLaMA-2-13B has the same performance for all quantized LoRA fine-tuning. I am curious regarding why this is the case?
* The authors metnion PiSSA and ApiQ in the background section. Clear comparison against these methods (even in the supplementary section) would make the claims of this paper even stronger.

**Methods And Evaluation Criteria:**

Yes, the methods and evaluation criteria are ideal for the application in hand. The metrics used are clearly defined and are commonly used in this setting.

**Other Comments Or Suggestions:**

N/A

**Other Strengths And Weaknesses:**

Strengths
* The problems of the current state of quantized LoRA fine tuning is clearly highlighted.
* The LowRA framework is clearly defined
* The ablation study is very helpful

Weaknesses
* It would help readability if the different components of the LowRA framework were explicitly mapped to the three limitations tackled in this paper.
* In many cases authors use data-free post-training which helps with the generalizability of the approach. However, for embedded systems fine-tuning with data can be helpful in some cases. I would like to know the authors thoughts in this aspect.

**Questions For Authors:**

N/A. I have already posed my questions in the previous sections.

**Relation To Broader Scientific Literature:**

The findings of this paper allows for LLMs to be adopted for systems with low memories such as embedded systems and mobile devices. This expands the use case of LLMs.

**Theoretical Claims:**

N/A. The paper does not make any theoretical claims.

---

> ### Author Rebuttal · Authors · 2025-04-01
>
> We thank reviewer BPFc for their insightful feedback! Below, we address the comments from the **Weaknesses** section in detail. We will incorporate your valuable feedback into the revised manuscript.
>
> ---
>
> ## Weakness 1: Clarity on LowRA’s Components
>
>
> > **Reviewer Concern**: The paper would benefit from more explicit mapping between the three limitations tackled and the different components of the LowRA framework.
>
> **Our Response**: We will clarify how each component of the LowRA framework aligns with each of the three limitations. Specifically:
>
> 1. **Limitation 1 (Coarse-Grained Precision Assignment)**
>    - **P2: Precision Assigner**
>    - **T3: Precs**
>
> 2. **Limitation 2 (Discrepancy in Data Distribution)**
>    - **P1: Mapping and Threshold Learner**
>    - **T2: Fine-Grained Mappings and Thresholds**
>    - **P4: Low-Rank Initializer** and **T5: Intelligently Initialized Low-Rank Tensors**
>      (the low-rank initialization helps mitigate quantization errors)
>
> 3. **Limitation 3 (Lack of High-Performance Quantization Primitives)**
>    - **P3: Output-Channelwise Quantize Kernel**
>    - **P5.1: Output-Channelwise Dequantize Kernel**
>
> We will include both graphical annotations and supporting text in the paper to further illustrate these mappings.
>
> ---
> ## Weakness 2: Fine-Tuning with Data in Embedded Systems
>
> > **Reviewer Concern**: Many approaches favor data-free post-training for generalizability. However, in embedded systems, fine-tuning with data can be more practical. How can LowRA adapt to such scenarios?
>
> **Our Response**: We interpret your comment as referring to scenarios where there is some prior knowledge of the tasks for which the LoRA base weight will be used. In embedded systems, these base weights might be:
>
>
> ### Case 1: Dedicated to a Single Task
>
> - **Using Task-Specific Data for Post-Training Quantization (PTQ).**
>   If the base weights are used solely for one task, that task’s dataset can serve as a calibration set to learn precision assignments, thresholds, mappings, and rounding schemes.
>
> - **Quantization-Aware Training (QAT).**
>   Another possibility is QAT. Preliminary experiments, however, showed that QAT did not outperform PTQ in terms of perplexity/accuracy—likely because LoRA’s low-rank adapters help compensate for quantization errors. Moreover, QAT demands more memory and introduces latency overhead.
>
> ### Case 2: Shared Across Multiple Tasks
>
> - **Representative Calibration Set for PTQ.**
>   If multiple tasks share certain characteristics, a representative calibration set may be synthesized for PTQ to learn shared precisions, thresholds, mappings, and rounding schemes.
>
> - **Possible QAT Approaches.**
>   While still possible, QAT in multi-task scenarios faces the same drawbacks regarding memory and computational overhead. Moreover, the learned precisions, thresholds, mapping, and rounding schemes may overfit the calibration set and fail to generalize to each individual downstream task.
>
> ### Other Scenarios and Future Directions
>
> - **Encoding vs. Decoding Schemes.**
>   One strategy is to fix encoding (thresholds) globally and fine-tune only the decoding (mappings) per task. However, preliminary implementations showed significant latency overhead (e.g., scatter-add operations).
>
> - **Future Extensions.**
>   We will continue exploring these ideas, focusing on deeper comparisons between PTQ and QAT under varying deployment constraints and use cases.
>
>
> ---
>
> ## Conclusion
>
> We appreciate your feedback and have outlined our approach to making LowRA clearer and more applicable to embedded scenarios. Your comments have inspired further investigations and future directions, which we look forward to sharing in subsequent work.

---

> > ### Comment · Reviewer_BPFc · 2025-04-04
> >
> > I thank the authors for their response. In particular, I appreciate them replying to my concerns. However, after going through the other reviews and comments I have decided not to update my score at this time.

---

> > > ### Author Response · Authors · 2025-04-05
> > >
> > > Dear Reviewer BPFc,
> > >
> > > We deeply thank you for your prompt and kind response.
> > >
> > > Thank you for your effort in putting together these valuable reviews. Your feedback has helped us tremendously. We unanimously agree that your observation on learning fine-grained quantization schemes with data for embedded settings opens up a lot of valuable research opportunities for us and the community at large.
> > >
> > > **In the following days, we would appreciate if you could let us know further questions you find interesting or important. We would try our best to address these questions.**
> > >
> > > We understand that it has been a lot of effort reading through the submissions and putting together reviews. We sincerely appreciate your effort.
> > >
> > > Hope you enjoy your days! Look forward to your response.
> > >
> > > Best,
> > >
> > > Submission13809 Authors

---

### Official Review · Reviewer_oGG6 · 2025-03-15

**Overall Recommendation:** 3

**Summary:**

This paper introduces LowRA, a novel framework that enables LoRA fine-tuning below 2 bits per parameter while maintaining model performance. The work addresses three key limitations of existing quantized LoRA methods through innovative techniques: fine-grained precision assignment, adaptive quantization mapping/thresholding, and efficient CUDA kernels for low-bit execution. The authors demonstrate significant memory savings (up to 50%) while achieving comparable or better performance compared to state-of-the-art methods.

## update after rebuttal

Thank you for the author's response. After referring to the author's response and the comments of other reviewers, I decided to keep my score.

**Claims And Evidence:**

The paper's claims are well-supported by comprehensive experimental results across multiple models (LLaMA-2-7B, 13B, 30B, BART-large) and tasks. The authors provide extensive ablation studies and detailed analysis of each component's contribution. The performance improvements and memory savings are clearly documented with quantitative metrics.

**Essential References Not Discussed:**

The paper adequately covers the relevant literature in LLM quantization and fine-tuning.

**Experimental Designs Or Analyses:**

The experimental evaluation is thorough and convincing:

1. Comprehensive baseline comparisons
2. Extensive ablation studies
3. Clear performance benchmarks
4. Detailed memory analysis
5. Multiple model scales tested

**Methods And Evaluation Criteria:**

The proposed methods are technically sound and well-motivated. The evaluation methodology is comprehensive, including:

1) Multiple model architectures and sizes
2) Various downstream tasks
3) Standard metrics (perplexity, ROUGE scores)
4) Detailed ablation studies
5) Memory usage analysis

**Other Comments Or Suggestions:**

None

**Other Strengths And Weaknesses:**

**Strengths:**

1. First framework to enable LoRA fine-tuning below 2 bits. Achieved 1.75 bits on LLaMA-2-7B/13B and BART-large, and 1.15 bits on LLaMA-30B with competitive performance. No previous methods support sub-2-bit operation.
2. Superior performance-precision trade-off. At 2-bit quantization, LowRA achieves perplexity reduction of 2.21 (WikiText-2) and 1.45 (Open-Assistant) over QLoRA, and 1.76 (WikiText-2) and 1.12 (Open-Assistant) over LoftQ.
3. Significant memory savings. Reduces memory usage by 30-50% compared to 4-bit baselines:
    - 40% lower memory for LLaMA-2-13B inference (2-bit vs 4-bit)
    - 30% lower for LLaMA-2-7B inference
    - 50% reduction for LLaMA-30B at 1.15 bits
4. Theoretically sound precision assignment strategy. Two-level ILP formulation with proven convergence, validated by ablation studies showing effectiveness of precision assigner.
5. Hardware-friendly design. Supports deployment on resource-constrained devices like Raspberry Pi 4 (4GB RAM) for LLaMA-2-7B and Tesla T4 (16GB VRAM) for LLaMA-30B.

**Weaknesses:**

1. Limited analysis of computational overhead during training. Paper focuses primarily on memory savings, with limited discussion of training time impacts in Section 7 and Appendix E.
2. Incomplete exploration of ultra-large models. Experiments stop at LLaMA-30B, without testing on larger models like LLaMA-65B or GPT-3 (175B).
3. Figure 10 shows variance analysis but lacks detailed explanation of impact on different layer types.
4. While training curves are shown, there's limited discussion of potential instability issues during ultra-low-bit training.
5. Impact on inference latency not thoroughly analyzed. Memory reduction is well-documented, but timing implications are not comprehensively measured and reported.

**Questions For Authors:**

Please refer to Strengths And Weaknesses.

**Relation To Broader Scientific Literature:**

The work builds upon and advances existing research in:

1. LoRA fine-tuning
2. Model quantization: QLoRA, LoftQ
3. Efficient LLM deployment
4. Low-bit neural networks

**Theoretical Claims:**

The theoretical foundation is solid, with clear mathematical formulations for:

1. Weighted Lloyd-Max quantization
2. Two-level ILP-based precision assignment
3. Channelwise precision optimization
4. Memory footprint analysis

---

> ### Author Rebuttal · Authors · 2025-04-01
>
> We thank reviewer oGG6 for their thorough review. We are glad that they find our work valuable overall. Below, we address the key concerns. We appreciate your feedback and will integrate it—along with additional experimental findings—into the revised manuscript.
>
> ---
>
> ## Training Overhead
>
> We benchmarked **LowRA** at various bit widths against **QLORA (4 bits)** on both an **A5000 (24GB VRAM)** and **A100 (80GB VRAM)**.
>
> ### Llama 7B A500, Runtime (ms)
> | Seq. Len. | Batch Size | 1.5 Bit  | 2.5 Bit  | 4 Bit    | QLORA (4 Bits) | Max % Overhead |
> |--------|-------|----------|-------|-------|------------|--------|
> | 256    | 1     | 782.67   | 789.37   | 782.67   | 754.27        | 4.65%          |
> | 512    | 1     | 1291.75  | 1295.82  | 1291.75  | 1268.50       | 2.15%          |
> | 1024   | 1     | 2398.10  | 2398.96  | 2398.10  | 2383.68       | 0.64%          |
> | 256    | 2     | 1268.17  | 1273.97  | 1268.17  | 1247.15       | 2.15%          |
> | 512    | 2     | 2319.83  | 2322.01  | 2319.83  | 2304.82       | 0.75%          |
>
>
> ### Llama 13B A5000, Runtime (ms)
> | Seq. Len.  | Batch Size | 1.5-bit  | 2.5-bit  | 3-bit    | 4-bit    | QLORA (4-bit) | Max % Overhead |
> |-----|-------|------|------|------|------|------|--------|
> | 256    | 1     | 1494.69  | 1494.69  | 1501.24  | 1508.03  | 1444.20       | 4.42%          |
> | 512    | 1     | 2486.00  | 2486.00  | OOM      | OOM      | OOM           | N/A            |
> | 256    | 2     | 2457.75  | 2457.75  | OOM      | OOM      | OOM           | N/A            |
>
> ### Llama 7B A100, Runtime (ms)
> | Seq. Len. | Batch Size | 1.5-bit  | 2.5-bit  | 3-bit    | 4-bit    | QLORA (4-bit) | Max % Overhead |
> |----|------|--------|-------|---|-----|-------|-----|
> | 256           | 1          | 632.09   | 632.36    | 631.85   | 639.31    | 589.12        | 8.52%          |
> | 512           | 1          | 1073.68   | 1074.60   | 1072.91  | 1079.48   | 1030.78       | 4.72%          |
> | 256           | 2          | 1056.94  | 1056.22   | 1055.86  | 1063.54   | 1014.91       | 4.79%          |
>
> LowRA introduces minimal overhead (with a maximum of 8.52%) and supports configurations **QLORA** cannot run without out-of-memory (OOM) errors.
>
>
> ---
>
> ## Results on Ultra-Large Models
> We evaluated **LowRA** on **LLaMA-65B** (1.15 bpp), achieving perplexities of **7.49** (WikiText) and **4.97** (OpenAssistant), outperforming **LLaMA-33B’s** **8.00** and **5.73** under identical hyperparameters. This confirms **LowRA’s** scalability to ultra-large models.
>
> ---
>
> ## Variance Trends Across Different Layer Types
> We will supplement our paper with fine-grained plots motivating output-channelwise quantization.
>
> ---
>
> ## Training Instability
> Even at 1.15 bits, we observed no instability or divergence, largely thanks to (1) LoftQ’s alternating SVD-based initialization of low-rank tensors and (2) freezing base weights while only updating low-rank tensors. We will include stability discussions in our paper.
>
> ---
>
> ## Inference implications
>
> **Inference implications involve three primary aspects: Loading Latency, Time to First Token (TTFT), and End-to-End Throughput.** We discuss each part here.
>
> 1. **Loading Latency**
>  This is the time to load model weights into GPU memory. On an RTX A4000 (LoRA branch in float32), LowRA reduces loading latency significantly, as shown in the table:
>
>    | Framework | Bits/Param | Loading Latency             |
>    |-----|------|-------|
>    | QLoRA     | 4          |  220.73 s                    |
>    | LowRA     | 4          | 218.17 s                    |
>    | LowRA     | 1.5        | 157.51 s (28.6% reduction)  |
>
> 2. **TTFT**
> TTFT is the delay before the first token is produced. While smaller models transfer less data and slightly reduce TTFT, it's primarily compute-bound, so quantization alone provides modest gains.
>
> 3. **End-to-End Throughput**
>     Once the model is loaded and past TTFT, throughput (tokens or requests per second) becomes the key factor.
>
>     ### Llama 7B on 1 RTX 3080 (unit: tokens / sec; float32)
>     |Prefill Length | Decode Length | 1.5 Bit | 2 Bit  | 2.5 Bit | 4 Bit  | QLoRA (4 Bit) |
>     |------|----|---|----|-------|-------|-----|
>     | 100            | 10            | 32.16   | 30.33  | 24.35   | 9.19   | 9.40          |
>
>
>     ### Llama 13B on 1 A4000 (unit: tokens / sec; float32)
>     | Prefill Length | Decode Length | 1.5 Bit | 2 Bit  | 2.5 Bit | 4 Bit  | QLoRA (4 Bit) |
>     |----|-----------|------|----|------|--------|-----|
>     | 100  |  10   |  16.11 | 14.46  | 13.64  |  12.17 |  11.56  |
>
>     Across **Llama 7B** on an **RTX 3080**, 1.5-bit LowRA delivers a **3.42× throughput increase** over QLoRA (32.16 vs. 9.40 tokens/sec), while on Llama 13B using an **RTX A4000**, **1.5-bit LowRA** still achieves a **1.39× speedup**—demonstrating that sub-2-bit quantization offers clear performance gains across varying model sizes.
>
> These results confirm that LowRA provides **real-world throughput gains beyond the documented memory savings**.

---

### Official Review · Reviewer_7mTX · 2025-03-24

**Overall Recommendation:** 3

**Summary:**

This paper introduces **LowRA**, a novel framework for LoRA-based fine-tuning of LLMs in ultra-low bit (sub-2-bit) settings. LowRA is the first to enable LoRA fine-tuning at or below 2 bits with only minor accuracy/perplexity losses and achieves considerable memory savings (30–50%).

The authors observe that current quantized LoRA methods (e.g. QLoRA, LoftQ) typically fail or degrade significantly below 2-4 bits. To address this, LowRA proposes a fine-grained, mixed-precision quantization strategy that - (a) learns quantization thresholds and representative values (mappings) via a weighted Lloyd-Max procedure, (b) uses a two-level ILP solver to assign channel-wise precisions, and (c) includes optimized low-bit CUDA kernels tailored to LoRA.

Experiments on Llama-2 (7B, 13B parameters) and BART-large, across tasks including WikiText-2, OpenAssistant, CNN/DailyMail, and XSum, with additional experiments on Llama 33B show that LowRA outperforms baselines in the performance-precision trade-off.

Notably, LowRA is the first method to enable Llama-30B fine-tuning on a single NVIDIA Tesla T4 (16GB VRAM).

**Claims And Evidence:**

- **Claim A:** LowRA can fine-tune LLMs below 2 bits (down to ~1.15 bits in some cases) without catastrophic performance degradation.
**Evidence:** Empirical results on Llama-2–7B and 13B (down to ~1.75 bits) [from Table 2] and Llama-33B (down to ~1.15 bits) [from Table 3] show perplexities close to baseline 2–4-bit methods.

- **Claim B:** LowRA yields superior or on-par performance compared to existing sub-4-bit baselines (QLoRA, LoftQ) while using fewer bits.
**Evidence:** On WikiText-2 and OpenAssistant, LowRA at ~2.5 bits approaches or exceeds the performance of QLoRA/LoftQ at 4 bits. [from Table 2]

- **Claim C:** LowRA substantially reduces memory usage (30–50%), valuable for resource-constrained fine-tuning and on-device inference.
**Evidence:** The authors detail memory estimates for Llama 2-7B,13B, & Llama-33B) in Appendix C.

**Essential References Not Discussed:**

None - to the best of my knowledge.

**Experimental Designs Or Analyses:**

The experimental designs and analyses seem fine.

**Methods And Evaluation Criteria:**

1. WikiText-2 (language modelling) and OpenAssistant (multi-turn conversation) are evaluated via perplexity.
2. XSUM and CNN/DailyMail (summarization tasks) are measured with ROUGE scores.

*Datasets make sense for the problem at hand.*
*Perplexity and ROUGE are among the most widely accepted metrics for these tasks. They ensure that both language generation quality (summarization) and text coherence/prediction (perplexity) are rigorously tested.*

**Other Comments Or Suggestions:**

Please see the word spacing for "Data-Free Post-Training Quantization". (~line 141)

Please fix the spelling error - "Quantzation". (Figure 5 caption)

Summarization Summarization (~line 699)

The mislabeled memory footprint of Llama 33B. (Figure 9)

Please refer to Meta's first generation of mid-tier model as Llama-33B (as was officially announced). There are numerous references to Llama-30B.

The right y-axis for the perplexity score exaggerates the drop visually. A full-scale axis would be a good choice, esp. for readers assessing quality vs. compression trade-offs. (Figure 7/8.)

In Algorithm 1, it would help to clarify the definitions of inputs like N, w(1),…,w(K). W_k and W_sum.

**Other Strengths And Weaknesses:**

**Strengths**
1. Empirical gains are convincing - particularly that performance remains stable even in ultra-low bit settings below 2 bits.
2. Overheads of the framework components are discussed. (Appendix E)
3. In the low bit range (2-4), the proposed framework outperforms LoftQ and QLora. (Table 2.)
4. The paper is easy to read and follow. Good overall presentation. esp. sections 5 and 6 - the core components of the paper.

**Weaknesses**
1. Although the framework supports sub-2-bit fine-tuning, insights into how sub-2-bit quantization affects LLM behaviour (e.g., interpretability, generalization) are limited. Some fundamental insight would be valuable, not just system performance.

2. While the paper makes progress in enabling ultra-low-bit LoRA fine-tuning and serving, it's not clear whether such ultra-low-bit quantized models outperform smaller models quantized at slightly higher bit widths. For instance, from Table 2, the **2-bit LowRA Llama-2-7B** (3808MB as from Figure 8) is comparable to the **1.8-bit LowRA Llama-2-13B** (5627MB as from Figure 8). Smaller models seem like a better choice than larger, more aggressively fine-tuned models esp. at lower bit widths scenarios. This raises an important question of whether it's always better to fine-tune larger models at lower precision, or whether smaller models at slightly higher precision might offer a better trade-off in both accuracy and throughput. A deeper discussion or analysis of this trade-off would strengthen the paper.

3. The motivation of the paper is ultra-low-bit fine-tuning for resource-constrained environments. Yet, all experiments were done on high-end A100. It would be valuable to include at least some inference-time measurements (e.g., latency, throughput, memory footprint) on representative low-resource hardware. This would help contextualize the practical system-level benefits and potential overheads of the proposed CUDA kernels and fine-grained quantization approach.

**Questions For Authors:**

In many real-world settings, the cumulative resources required for inference in deployment far exceed the one-time costs of fine-tuning.

**Q1.** Do you have inference throughput benchmarks (e.g., tok/sec) for inference with and without LowRA’s sub-2-bit quantization? This would clarify real-world gains beyond memory savings.

**Q2.** Can ultra-low-bit quantization with LowRA justify choosing a larger model (e.g., 13B @ 1.5 bits) over a smaller 7B model at 2–4 bits (with LoftQ)? Any guidance on this trade-off? Thanks.

**Relation To Broader Scientific Literature:**

LowRA builds directly on the concept of parameter-efficient fine-tuning (PEFT), esp. LoRA (Hu et al., 2021), and extends quantized LoRA approaches like QLoRA and LoftQ. It adopts groupwise quantization ideas from prior work (e.g., GroupWise Normalization in QLoRA), but pushes further by combining a Weighted Lloyd-Max algorithm for adaptive thresholding with two-step ILP-based precision assignment, enabling sub-2-bit training. This unified approach addresses limitations observed in earlier methods (which were mostly restricted to 2-4 bits) and draws on established mixed precision and quantization techniques.

**Theoretical Claims:**

The main theoretical component is the ILP-based assignment of channel-wise precisions. While no formal proof of optimality is given for the entire method’s synergy with LoRA - the authors do provide a well-defined ILP objective and a hierarchical solution scheme. The Weighted Lloyd-Max approach is grounded in standard quantization theory. Everything is consistent with known quantization methods; there are no obviously incorrect proofs. The approach is algorithmic rather than strictly theoretical, and the correctness of the optimization steps seems reasonable.

---

> ### Author Rebuttal · Authors · 2025-04-01
>
> We thank reviewer 7mTX for their thorough and insightful review. We are glad that 7mTX finds our empirical gains convincing and our paper easy to read and follow. We address all feedback below and will incorporate it, along with new experimental findings, into the revised paper.
>
> ---
>
> ## Q1: Inference metrics on consumer-grade hardware
>
> Our experiments target GPUs that are relatively constrained for large language model (LLM) inference—an **NVIDIA RTX 3080** with **10GB VRAM** and an **NVIDIA RTX A4000** with **16GB VRAM**—to underscore how LowRA can unlock efficient inference without requiring high-end data-center hardware.
>
>
> ### Llama 7B on RTX 3080 (unit: tokens / sec; float32)
> | Metric        | Prefill Length | Decode Length | 1.5 Bit | 2 Bit  | 2.5 Bit | 4 Bit  | QLoRA (4 Bit) |
> |---------|----------------|---------------|---------|--------|---------|--------|---------|
> | **Throughput**| 100            | 10            | 32.16   | 30.33  | 24.35   | 9.19   | 9.40          |
> | **Speedup**   | 100            | 10            | 3.42x    | 3.23x   | 2.59x    | 0.98x   | 1.00x          |
>
>  - At **1.5 bits**, throughput reaches **32.16 tokens/sec**, yielding a **3.42× speedup** over QLoRA (9.40 tokens/sec).
>  - Even at **2 bits**, LowRA surpasses QLoRA by over **3×**, indicating that more aggressive quantization pays off on memory-limited hardware.
>
>
>
> ### Llama 13B on A4000 (unit: tokens / sec; float32)
> | Metric        | Prefill Length | Decode Length | 1.5 Bit | 2 Bit  | 2.5 Bit | 4 Bit  | QLoRA (4 Bit) |
> |---------------|----------------|---------------|---------|--------|---------|--------|---------------|
> | **Throughput**|  100  |  10   |  16.11 | 14.46  | 13.64  |  12.17 |  11.56  |
> | **Speedup**   | 100           | 10            | 1.39x    | 1.25x   | 1.18x    | 1.05x   | 1.00x          |
>
>
> - For **Llama 13B**, 1.5-bit LowRA achieves **1.39×** speedup over QLoRA, and 2-bit still delivers a **1.25×** improvement.
>  - Though the gains are smaller than on the 7B model—likely due to higher overall computation overhead—LowRA still outperforms QLoRA across all tested bit-widths.
>
> These results confirm that our sub-2-bit LowRA quantization provides real-world throughput gains beyond the documented memory savings.
>
> ---
>
> ## Q2: The tradeoff of model size and compression ratio.
> In practice, users often start with a specific model, a fixed hardware memory budget, and particular tasks in mind. LowRA is designed to help maximize performance under those given constraints.
>
> While we do observe that ultra-low-bit quantization can make it feasible to run larger models within limited memory, our current study does not fully investigate the potential trade-off between choosing a bigger model at extremely low bits (e.g., 13B at 1.5 bits) and opting for a smaller model at higher bits (e.g., 7B at 2-4 bits).
>
> We see further exploration of this bits-per-parameter versus performance trade-off - along with automatic selection of the optimal model-compression combination - as an important direction for future research. Moreover, our work serves as a stepstone for future works to further optimize the task performance of models in the lower bit range.
>
>
> ---
>
> ## Presentation Errors: Graphs, Text, and Algorithms
>
> Thank you for your attention to details! We have fixed all of the presentation errors you pointed out. We will iterate through our draft rigorously to ensure our presentation is accurate.
>
> 1. **Typographical and Spacing Fixes**
>    - Corrected spacing for “Data-Free Post-Training Quantization.”
>    - Fixed spelling errors (e.g., “Quantzation” → “Quantization,” “Summarization Summarization” → “Summarization”).
>
> 2. **Figure and Label Updates**
>    - Corrected the mislabeled memory footprint for Llama 33B in Figure 9.
>    - Standardized references to “Llama-33B” (instead of “Llama-30B”).
>    - Adjusted the y-axis to a full-scale view in Figures 7 and 8, ensuring a clearer comparison of perplexity scores.
>
> 3. **Algorithm Clarifications**
>
>     We have added clarifications to the algorithm. Specifically, $N$ refers to the total number of output channels in the model being quantized. The values $\{ w^{(1)}, \dots, w^{(K)} \}$ represent the distinct parameter sizes (i.e., the number of parameters) across all output channels. The set $I_k$ contains the channel indices belonging to size group \(k\). Each channel \(i \in I_k\) has a parameter count of $w^{(k)}$.
>
>     We define:
>
>     $$
>     W_k = \sum_{i \in I_k} w^{(k)},
>     $$
>
>     which is the sum of parameter counts for all channels in partition $I_k$. Finally:
>
>     $$
>     W_{\text{sum}} = \sum_{k=1}^K W_k
>     $$
>
>     is the total parameter count across all partitions (i.e., the total number of parameters in the model to be quantized).

---

> > ### Comment · Reviewer_7mTX · 2025-04-04
> >
> > I sincerely thank the authors for their time and effort in addressing my questions. particularly for sharing the inference-related implications.
> >
> > LowRA unlocks more quantization choices (below 2 bits) for LLMs - which naturally raises this question of finding the optimal model-compression combination. While this tradeoff does not fall within the scope of this paper, I take it as a positive sign that more works could be built on this paper to explore more in this domain.
> >
> > I will update my recommendation to 3.

---

> > > ### Author Response · Authors · 2025-04-05
> > >
> > > Dear Reviewer 7mTX,
> > >
> > > Thank you for your prompt response!
> > >
> > > It is our pleasure to have this submission reviewed by you. Your sharp and detailed feedback has helped us crucially in furthering and enhancing this line of research. In particular, your question on the inference metrics has prompted us better complete our work in incorporating aspects of system performance. Moreover, your astute consideration on the tradeoff between original model size and compression ratio has pointed us (and the quantization community) to a new/yet-to-be-investigated topic of research.
> > >
> > > **In the coming days, we would like to address any further questions/concerns you may have. We would try our best to enhance our work to be better received by the community.**
> > >
> > > We understand it is a lot of effort putting together these valuable reviews that attend to so many details. We sincerely appreciate your effort in cultivating a healthy, mutually-motivating community.
> > >
> > > Best wishes,
> > >
> > > Submission13809 Authors

---

### Decision · Program_Chairs · 2025-05-01

**Decision:**

Accept (poster)

**Comment:**

All three reviewers recommended to accept this paper in their reviews. After discussion, it was eventually agreed that this paper meets the ICML acceptance bar.
The authors are encouraged to address the comments of the reviewers to improve this work.
The author’s rebuttal has been carefully read and discussed. The author’s message has been carefully read and considered.